# Every Rollout Counts: Optimal Resource Allocation for Efficient Test-Time Scaling

**Xinglin Wang**[1], **Yiwei Li**[1], **Shaoxiong Feng**[2][†] **Peiwen Yuan**[1], **Yueqi Zhang**[1],
**Jiayi Shi**[1], **Chuyi Tan**[1], **Boyuan Pan**[2], **Yao Hu**[2], **Kan Li**[1][†]

[1] School of Computer Science, Beijing Institute of Technology
[2] Xiaohongshu Inc

{wangxinglin,liyiwei,peiwenyuan,zhangyq,shijiayi,tanchuyi,likan}@bit.edu.cn
{shaoxiongfeng2023}@gmail.com {panboyuan,xiahou}@xiaohongshu.com

## Abstract

Test-Time Scaling (TTS) improves the performance of Large Language Models (LLMs) by using additional inference-time computation to explore multiple reasoning paths through search. Yet how to allocate a fixed rollout budget most effectively during search remains underexplored, often resulting in inefficient use of compute at test time. To bridge this gap, we formulate test-time search as a resource allocation problem and derive the optimal allocation strategy that maximizes the probability of obtaining a correct solution under a fixed rollout budget. Within this formulation, we reveal a core limitation of existing search methods: solution-level allocation tends to favor reasoning directions with more candidates, leading to theoretically suboptimal and inefficient use of compute. To address this, we propose Direction-Oriented Resource Allocation (DORA), a provably optimal method that mitigates this bias by decoupling direction quality from candidate count and allocating resources at the direction level. To demonstrate DORA's effectiveness, we conduct extensive experiments on challenging mathematical reasoning benchmarks including MATH500, AIME2024, and AIME2025. The empirical results show that DORA consistently outperforms strong baselines with comparable computational cost, achieving state-of-the-art accuracy. We hope our findings contribute to a broader understanding of optimal TTS for LLMs.[1]

## 1 Introduction

As the challenges of scaling up computation and data resources for pretraining continue to grow, scaling test-time computation has emerged as a critical paradigm for enhancing model performance (Brown et al., 2024; Snell et al., 2024; Wu et al., 2025a). By allocating additional computation at inference time, Test-Time Scaling (TTS) improves the performance of LLMs on complex tasks such as mathematical reasoning by enabling deeper exploration of possible solutions (Qwen Team, 2024; Kimi Team et al., 2025; DeepSeek-AI et al., 2025). One prominent approach to scaling test-time computation is through search, where diverse candidate solutions are proposed and filtered using a Process Reward Model (PRM) to guide the procedure (Chen et al., 2024b; Snell et al., 2024; Beeching et al., 2024; Wu et al., 2025a; Liu et al., 2025). By pruning low-quality paths early and focusing computation on more promising ones, these strategies help steer the search process toward trajectories that are more likely to yield correct answers (Setlur et al., 2025b).

While these strategies yield promising performance gains, the question of how to optimally allocate a fixed rollout budget across competing candidate trajectories remains underexplored. In practice,

---

[†]Corresponding author.
[1]Our code and data have been released on `https://github.com/WangXinglin/DORA`.

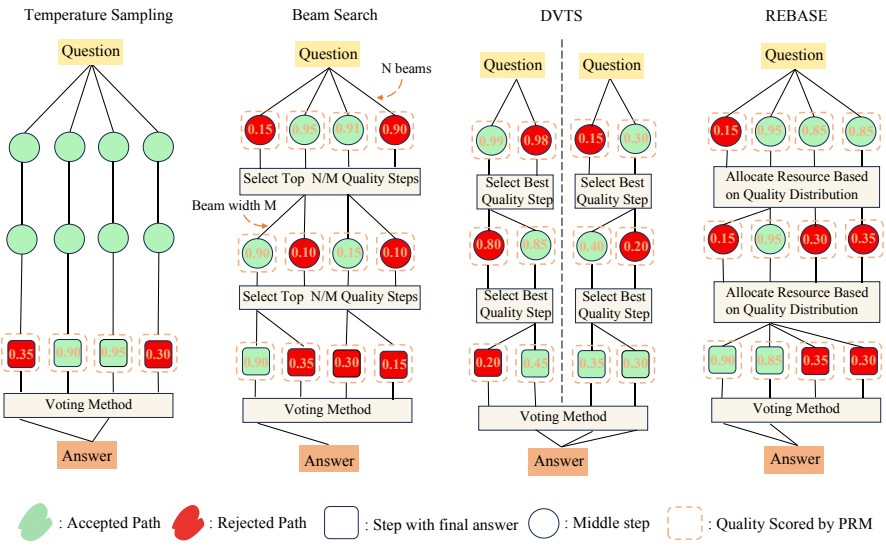

Figure 1: Comparison of different parallel Test-Time search strategies.

existing strategies rely on human-designed heuristics (Figure 1): preserving certain number of high-quality candidates (Beam Search) (Snell et al., 2024), promoting diversity (DVTS) (Beeching et al., 2024), or balancing exploration and exploitation (REBASE) (Wu et al., 2025a). While these intuitions offer practical value, they lack a principled foundation and do not provide guarantees of optimality, such as maximizing the probability of obtaining a correct solution. As a result, rollout budgets may be allocated inefficiently, limiting the effectiveness of test-time computation.

To bridge this gap, we formulate test-time search as a resource allocation problem, where the goal is to maximize the probability of obtaining a correct solution under a fixed rollout budget (Section 3.1). Based on this formulation, we derive the theoretical form of the optimal allocation strategy and revisit existing search methods through a unified lens. We show that, under the assumption that candidate solutions are independent, several widely used strategies approximate the optimal allocation corresponding to different assumptions about the reliability of the reward estimates. However, this independence assumption does not hold in practice, as many candidates share the same underlying reasoning direction (Bi et al., 2024; Hooper et al., 2025). Our theoretical analysis further shows that solution-level allocation is suboptimal: it conflates direction quality with candidate count, biasing the allocation toward overrepresented directions and leading to inefficient use of test-time compute (Section 3.2).

To address this issue, we propose Direction-Oriented Resource Allocation (DORA), a provably optimal method that corrects for this allocating bias by decoupling direction quality from candidate count and allocating resources at the direction level. To validate the effectiveness of DORA, we evaluate it on the challenging mathematical benchmarks MATH500 (Hendrycks et al., 2021), AIME2024 (AI-MO, 2024), and AIME2025 across a broad range of rollout budgets and policy models. The empirical results show that DORA consistently outperforms strong baseline strategies under comparable computational budgets, highlighting its ability to improve the effectiveness of each rollout and enhance the overall efficiency of TTS.

## 2 Setup & Preliminaries

### 2.1 Problem Formulation

We formulate the parallel search process under a unified framework, defined by the tuple $(\pi, Q, O, V, N)$, where $\pi(a \mid \tau)$ is a policy model that generates an action $a$ (reasoning step) given a partial solution $\tau = (x, a_1, \ldots, a_i)$, where $x$ denotes the input problem; $Q : \tau \mapsto [0, 1]$ is the Process Reward Model (PRM), which scores the quality of a partial or complete solution; $O : \mathbb{R}^N \to \mathbb{N}_+^N$ is the resource allocation strategy, dynamically assigning computational budget based on solution

scores; $V$ is the voting method that aggregates final answers from completed solutions to select the most likely correct final answer (e.g., via majority voting, best-of-N, or weighted best-of-N); and $N$ is the total rollout budget of parallel explorations.

The parallel search process can be summarized as Algorithm 1. Specifically, the process iteratively expands a set of partial solutions using the policy $\pi$, collects complete solutions, and redistributes the rollout budget via the allocation strategy $O$ based on intermediate rewards from $Q$. Once sufficient complete solutions are gathered, the final answer is selected using the voting method $V$.

## 2.2  Parallel Search Method

We consider four parallel TTS methods which are popularly used in practice: Temperature Sampling (Brown et al., 2024), Beam Search (Snell et al., 2024), Diverse Verifier Tree Search (DVTS) (Beeching et al., 2024), and Reward Balanced Search (REBASE) (Wu et al., 2025a). As pointed out by Snell et al. (2024), lookahead search is inefficient due to sequential sampling, so we do not include it or other methods involving lookahead operations, such as Monte Carlo Tree Search (MCTS).

Based on the unified framework above, we now analyze these strategies from the perspective of resource allocation. While sharing the same overall structure, they differ solely in their choice of allocation function $O(\boldsymbol{R})$, which determines how the total rollout budget $N$ is distributed across candidate solutions based on their PRM scores. We denote the number of rollouts assigned to the $i$-th candidate $\tau_j$ as $O(\boldsymbol{R})_i$, where $O$ is the allocation function and $\boldsymbol{R} = \{R_1, \ldots, R_k\}$ is the vector of PRM scores.

**Temperature Sampling.**  This method performs sampling purely from the policy model, without using reward information for rollout allocation. All candidates are treated equally, and each receives one rollout. External reward signals may still be used at the final answer selection stage, e.g., through best-of-$N$ or weighted best-of-$N$ voting.

$$O_{\text{Temp}}(\boldsymbol{R})_i = 1. \tag{1}$$

**Beam Search.**  Beam Search selects the top $K = N/M$ candidates based on their PRM scores, where $M$ is the number of rollouts assigned per candidate (i.e., the beam width). Only the top-$K$ receive any rollout allocation, while the rest are discarded:

$$O_{\text{Beam}}(\boldsymbol{R})_i = \begin{cases} M, & \text{if } i \in \text{Top-}K(\boldsymbol{R}), \\ 0, & \text{otherwise.} \end{cases} \tag{2}$$

**DVTS.**  To encourage exploration across diverse solution branches, DVTS partitions the $k$ candidates into $K = N/M$ disjoint groups of size $M$, corresponding to independent subtrees. Within each group, it performs a local Beam Search by selecting the candidate with the highest PRM score and assigning it $M$ rollouts. Only one candidate per group receives any resource, and groups do not share information:

$$O_{\text{DVTS}}(\boldsymbol{R})_i = \begin{cases} M, & \text{if } i = \arg\max_{j \in \mathcal{G}(i)} R_j, \\ 0, & \text{otherwise,} \end{cases} \tag{3}$$

where $\mathcal{G}(i)$ denotes the group containing candidate $i$.

**REBASE.**  Instead of selecting a fixed number of candidates, REBASE distributes the total rollout budget more smoothly based on the relative quality of each candidate to balance exploitation and exploration. It applies a softmax over the PRM scores $R_i$ to compute allocation weights, and assigns rollouts proportionally:

$$O_{\text{REBASE}}(\boldsymbol{R})_i = \text{round}\left(N \cdot w_i\right), \quad \text{where } w_i = \frac{e^{R_i/T_b}}{\sum_j e^{R_j/T_b}}. \tag{4}$$

where $T_b$ is a temperature parameter controlling the sharpness of the allocation.

# 3 Optimal Parallel Search for Test-Time Scaling

While previous parallel search methods enable efficient TTS by exploring multiple reasoning paths simultaneously, their effectiveness critically depends on how the fixed compute budget (i.e., number of rollouts) is allocated across candidate solutions. We focus on the following question:

> *Given a fixed rollout budget, how should one allocate resources across candidate reasoning paths to maximize performance (i.e., the success rate of achieving a correct solution)?*

We are the first to formulate this problem and study the associated parallel search strategies, setting our work apart from previous parallel search studies (Wu et al., 2025a; Beeching et al., 2024; Jiang et al., 2024). To address this, we introduce a Bayesian probabilistic model of solution correctness, and derive an allocation strategy that maximizes expected success under a rollout budget constraint.

## 3.1 Theoretical Formulation of Optimal Resource Allocation

We aim to allocate a fixed rollout budget $N$ across $k$ candidate reasoning paths to maximize the probability of solving the problem correctly, i.e., obtaining at least one successful solution. Let $p_i \in [0, 1]$ denote the (unknown) success probability of the $i$-th candidate $\tau_i$ when sampled once.

**Assumption 1.** *The success events of different candidate solutions are independent.*

Under Assumption 1, the probability of obtaining at least one success under an allocation vector $\boldsymbol{B} = \{B_i\}_{i=1}^k$ is given by:

$$\mathbb{P}(\text{success}) = 1 - \prod_{i=1}^{k}(1 - p_i)^{B_i}. \tag{5}$$

Since the true values of $p_i$ are unknown, we adopt a Bayesian modeling approach to capture the uncertainty in their estimation. In practice, $p_i$ is often approximated using the Process Reward Model (PRM) score $R_i = Q(\tau_i)$ (Wang et al., 2024a; Luo et al., 2024; Wang et al., 2024b; Setlur et al., 2025a; Lee et al., 2025), which serves as a proxy for the probability of correctness. However, these estimates are subject to considerable noise due to imperfections in the policy model, variations in decoding temperature, and inherent sampling randomness. To model this uncertainty explicitly, we treat each $p_i$ as a latent variable and place a Beta prior over it. Specifically, we normalize the PRM score into $w_i \in (0, 1)$, and define:

$$p_i \sim \text{Beta}(\kappa w_i, \kappa(1 - w_i)), \tag{6}$$

where $\kappa > 0$ controls the concentration of the prior around its mean. Larger values of $\kappa$ correspond to higher confidence in the PRM estimate $w_i$, while smaller values encode greater uncertainty (see Appendix C for more details).

Our goal is to maximize the probability of obtaining at least one successful solution. Under the Bayesian model, this is equivalent to minimizing the expected joint failure:

$$\min_{\sum B_i = N} \mathbb{E}\left[\prod_{i=1}^{k}(1 - p_i)^{B_i}\right]. \tag{7}$$

This defines a convex optimization problem over the rollout allocation vector $\boldsymbol{B} = \{B_i\}_{i=1}^k$. By applying the Karush-Kuhn-Tucker (KKT) conditions, we characterize the limiting behavior of the optimal allocation (see Appendix B.1 for details of proof):

**Proposition 1** (Limiting Behavior of Optimal Allocation). *Let $O^\star(\boldsymbol{w})_i$ denote the optimal rollout allocation for candidate $i$, where $\boldsymbol{w} = \{w_1, \ldots, w_k\}$ are the normalized PRM scores. Then:*

- *When $\kappa \to 0$, the optimal allocation assigns one rollout to each of the top-$\min(k, N)$ candidates with highest $w_i$ scores:*

$$O^\star(\boldsymbol{w})_i = \begin{cases} 1, & \text{if } i \in \text{Top-}\min(k, N) \text{ of } \boldsymbol{w}, \\ 0, & \text{otherwise}, \end{cases}$$

*with the remaining $N - \min(k, N)$ rollouts arbitrarily assigned.*

- *When $\kappa \to \infty$, the optimal allocation converges to a deterministic allocation that assigns all rollouts to the highest-scoring candidate:*

$$O^\star(\boldsymbol{w})_i = \begin{cases} N, & \text{if } i = \arg\max_j w_j, \\ 0, & \text{otherwise.} \end{cases}$$

- *When $\kappa$ is fixed and finite, the optimal allocation approximately follows a shifted linear rule:*

$$O^\star(\boldsymbol{w})_i \approx (N + k\kappa) \cdot w_i - \kappa.$$

Proposition 1 shows that the optimal allocation strategy evolves continuously with the confidence parameter $\kappa$. When $\kappa \to \infty$, the Beta prior becomes highly concentrated around the PRM estimate $w_i$, reflecting strong confidence in its accuracy. In this case, the optimal solution assigns the entire rollout budget to the top-ranked candidate, effectively recovering Beam Search with beam width $M = N$ (Equation 2) and fully exploiting the highest-scoring path.

Conversely, when $\kappa \to 0$, the Beta prior becomes maximally uncertain, collapsing to a Bernoulli mixture where each candidate has a binary chance of being correct or incorrect, with prior weight $w_i$. In this setting, relying heavily on any single PRM estimate becomes risky, as the scores provide no meaningful guidance. To mitigate this risk, the optimal strategy spreads the rollout budget across multiple candidates in proportion to their prior likelihoods. This reduces to sampling top candidates according to a multinomial distribution over $w_i$, a behavior closely aligned with temperature sampling used in stochastic decoding.

When $\kappa$ is fixed and finite, the optimal allocation takes a smoothed, uncertainty-aware form that interpolates between the two extremes above. Specifically, the rollout budget is approximately distributed according to a shifted linear rule (Proposition 1), which closely matches the REBASE strategy (Equation 4). In this regime, the PRM scores are treated as informative but noisy, and the allocation strategy balances exploration and exploitation accordingly.

In practice, due to sampling noise and imperfections in the policy model, PRM scores carry considerable uncertainty. Consistent with this observation, we find that REBASE, which allocates rollouts in proportion to PRM scores, outperforms alternative strategies across a wide range of tasks (see Figure 3). This supports the relevance of the $\kappa \to 0$ setting, which we adopt as the default throughout the paper. Accordingly, we treat REBASE as the baseline solution-level allocation strategy in all subsequent analysis.

## 3.2 Suboptimality of Solution-Level Allocation

While REBASE is optimal under the assumption of candidate independence (Assumption 1), this condition often does not hold in practice. In particular, many candidate solutions share the same underlying reasoning direction (Bi et al., 2024; Hooper et al., 2025), forming clusters of highly correlated outputs. The solution-level nature of REBASE leads to skewed allocation when candidate counts are imbalanced across reasoning directions.

To formalize this issue, we group candidate solutions into $g$ reasoning directions. Let direction $j$ contain $k_j$ candidates, all sharing the same PRM score $R_j$, and let $\mathcal{E}_j$ denote the index set of these candidates.

Under REBASE, rollout allocation is performed at the solution level, which implicitly induces a direction-level allocation according to Eq. 4:

$$B_j^{(\text{solution})} = \sum_{i \in \mathcal{E}_j} N \cdot \frac{e^{R_j}}{\sum_{l=1}^g k_l e^{R_l}} = N \cdot \frac{k_j e^{R_j}}{\sum_{l=1}^g k_l e^{R_l}}. \tag{8}$$

In contrast, the optimal allocation strategy would treat each reasoning direction as a single unit and assign rollouts in proportion to the softmax over direction-level scores:

$$B_j^{(\text{direction})} = N \cdot \frac{e^{R_j}}{\sum_{l=1}^g e^{R_l}}. \tag{9}$$

By comparing the induced solution-level allocation in Eq. 8 with the optimal direction-level allocation in Eq. 9, we derive the following proposition (see Appendix B.2 for details of proof):

**Proposition 2** (Suboptimality of Solution-Level Allocation). *When Assumption 1 does not hold, the solution-level allocation $B_j^{(solution)}$ is suboptimal: it does not match the optimal direction-level allocation $B_j^{(direction)}$ unless all directions contain the same number of candidate solutions, i.e., $k_j = k$ for all $j$.*

This result reveals a fundamental limitation of solution-level allocation: it implicitly favors reasoning directions with more candidate solutions (Figure 2). This bias results in inefficient use of the rollout budget, motivating our proposed method: Direction-Oriented Resource Allocation (DORA).

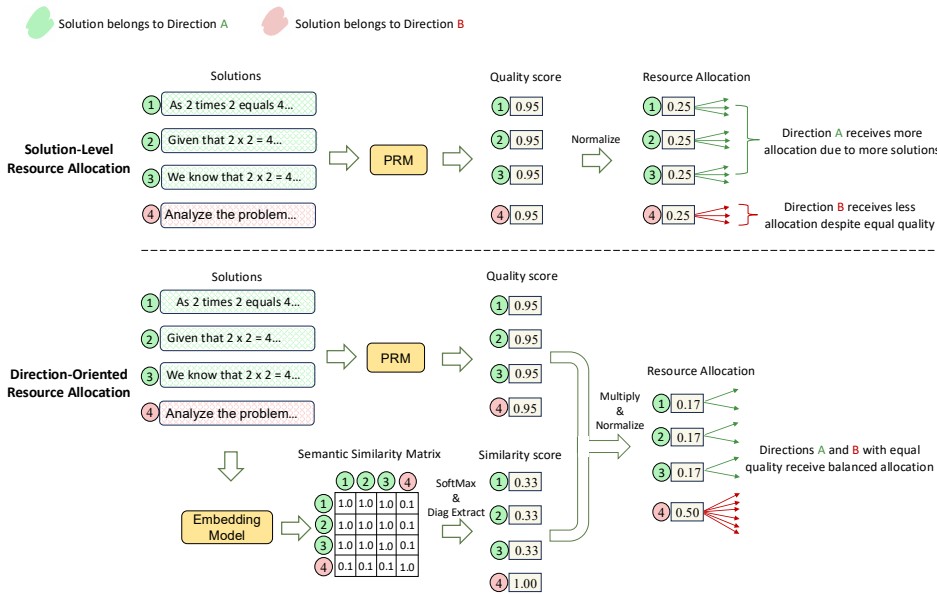

Figure 2: Comparison between Solution-Level Resource Allocation and proposed Direction-Oriented Resource Allocation (DORA).

## 3.3 Direction-Oriented Resource Allocation (DORA)

To address the bias introduced by solution-level allocation, we propose DORA, a method that adjusts rollout allocation by identifying and correcting for structural redundancy among candidate solutions. As illustrated in Figure 2, DORA incorporates semantic structure into the allocation process by softly clustering solutions into shared reasoning directions and assigning rollouts proportionally at the direction level, rather than treating each solution independently.

Given a set of candidate solutions $\{\tau_1, \ldots, \tau_k\}$, DORA first estimates which solutions share reasoning structure by computing semantic embeddings $\boldsymbol{e}_i \in \mathbb{R}^d$ via a pretrained embedding model. These embeddings are used to construct a cosine similarity matrix $S \in \mathbb{R}^{k \times k}$:

$$S_{ij} = \frac{\boldsymbol{e}_i^\top \boldsymbol{e}_j}{\|\boldsymbol{e}_i\| \cdot \|\boldsymbol{e}_j\|}. \tag{10}$$

To avoid hard clustering and retain flexibility, we interpret the similarity between candidates as a soft assignment over directions. Specifically, we apply a row-wise softmax over $S$ with temperature $T_s$, yielding an affinity matrix $P \in \mathbb{R}^{k \times k}$:

$$P_{ij} = \frac{e^{S_{ij}/T_s}}{\sum_{j'=1}^{k} e^{S_{ij'}/T_s}}. \tag{11}$$

The diagonal entry $\gamma_i = P_{ii}$ then measures the *semantic uniqueness* of solution $\tau_i$, serving as a proxy for the inverse size of the solution's underlying direction.

Following the REBASE formulation in Eq. 4, we compute normalized quality weights $w_i$ from PRM scores $R_i = Q(\tau_i)$ using a softmax with temperature $T_b$.

To incorporate semantic structure, we reweight each $w_i$ by its uniqueness:

$$w_i' = \frac{w_i \cdot \gamma_i}{\sum_{j=1}^{k} w_j \cdot \gamma_j}. \tag{12}$$

This downweights redundant solutions and redistributes resources toward distinct reasoning directions.

Finally, rollouts are allocated proportionally:

$$B_i = \text{round}(N \cdot w_i'). \tag{13}$$

DORA balances rollouts across semantically distinct reasoning directions, mitigating the redundancy bias of solution-level methods like REBASE. As summarized in Theorem 1, DORA yields the optimal direction-level allocation under mild assumptions (See Appendix B.3 for the full derivation).

**Theorem 1** (Optimality of DORA). *Assume candidate solutions are grouped into $g$ reasoning directions, where direction $j$ consists of candidates indexed by $\mathcal{E}_j$, and all candidates in $\mathcal{E}_j$ share the same PRM score $R_j$. Then DORA recovers the optimal direction-level rollout allocation specified in Eq. 9.*

## 4 Experiments

### 4.1 Experimental Setup

We use Qwen2.5-Math-PRM-7B (Zhang et al., 2025) as our Process Reward Model (PRM) due to its superior reward estimation performance (Zheng et al., 2024; Song et al., 2025). For the policy models, we include Llama-3.2-1B-Instruct, Llama-3.2-3B-Instruct (AI, 2024), and Qwen2.5-1.5B-Instruct (Yang et al., 2024), covering a range of model scales and architectures. Considering that existing open-source PRMs are primarily trained on mathematical tasks, we focus our evaluation on three challenging math reasoning benchmarks: MATH500, AIME2024, and AIME2025. To provide a more comprehensive assessment of reasoning performance, we further include four additional math reasoning benchmarks: HMMT24, HMMT25, AMC23, and AMC24, which feature diverse question formats and difficulty distributions. We evaluate models under rollout budgets of 16, 32, 64, 128, and 256 on the main benchmarks. Following Hochlehnert et al. (2025), we repeat all experiments five times on MATH500 and ten times on AIME2024 and AIME2025, reporting the average performance across all runs to reduce the impact of randomness and improve the reliability of our conclusions. For reward assignment during rollouts, we use the final PRM score at each step as the reward for that step. The final answer is selected using weighted majority voting, where each trajectory is weighted by its final PRM score. We use these aggregation strategies since they have been shown to outperform other methods of aggregating trajectories to determine the final response (Beeching et al., 2024). See Appendix E.1 for experimental hyperparameters.

### 4.2 Main Results

**DORA is the most effective parallel search method.** As shown in Figure 3 (a) and Table 1, DORA consistently achieves the highest accuracy across all policy models on MATH500, AIME2024, AIME2025, and four additional math reasoning datasets (HMMT24, HMMT25, AMC23, and AMC24). This consistent superiority demonstrates DORA's advantage to make more efficient use of limited test-time compute compared to baseline strategies. To better understand this advantage, we further analyze the pass rate (the number of correct solutions among all sampled rollouts). As shown in Figure 3 (b), DORA consistently reaches more correct solutions than other baselines, highlighting its effectiveness in exploring a broader set of high-quality reasoning paths. Notably, the performance gap between DORA and REBASE widens as the rollout budget increases. We hypothesize that this is due to growing redundancy in sampled solutions: with more rollouts, a larger proportion of trajectories tend to converge to similar final solutions, making REBASE's solution-oriented allocation increasingly prone to overestimating certain reasoning directions. In contrast, DORA mitigates this issue by allocating rollouts at the direction level, allowing for more accurate resource allocation.

### 4.3 Analysis

**DORA is compute-optimal.** Considering that DORA introduces an additional semantic similarity step via an embedding model, we examine whether the associated computational overhead is justified

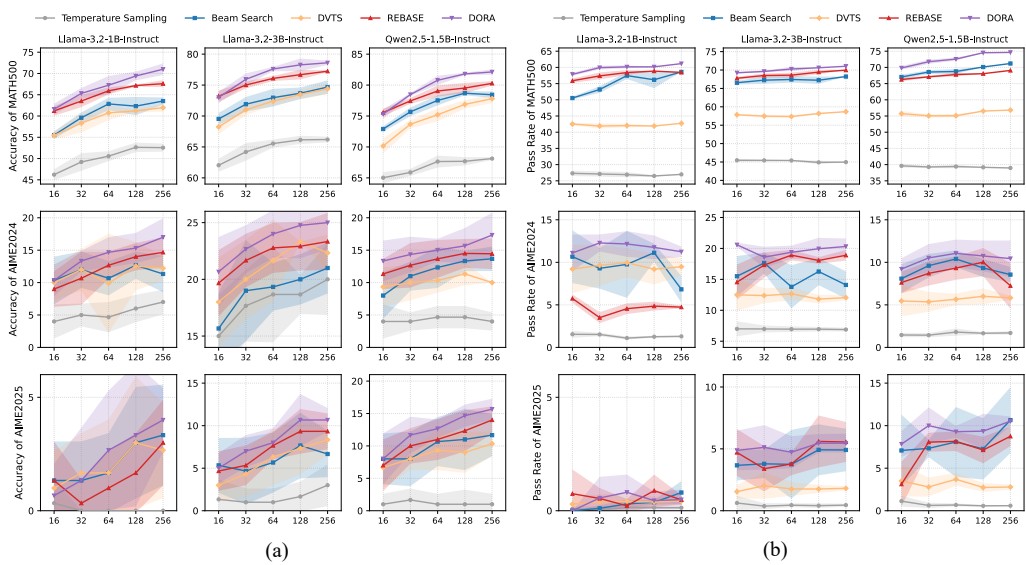

Figure 3: Accuracy and Pass rate comparison under various rollout budgets on MATH500, AIME2024, and AIME2025.

Table 1: Performance comparison on broader math reasoning benchmarks. We set rollout budget as 64, and all results are averaged over five runs.

| Policy Model | Method | HMMT24 | HMMT25 | AMC23 | AMC24 |
|---|---|---|---|---|---|
| LLaMA-3.2-1B-Instruct | Temperature Sampling | 0.0 | 0.0 | 28.0 | 13.3 |
| | Beam Search | 2.0 | 0.0 | 41.5 | 15.5 |
| | DVTS | 1.3 | 0.0 | 41.0 | 18.0 |
| | REBASE | 2.0 | 0.0 | 45.0 | 20.0 |
| | DORA | **3.3** | 0.0 | **45.0** | **20.0** |
| LLaMA-3.2-3B-Instruct | Temperature Sampling | 0.7 | 0.0 | 48.5 | 21.7 |
| | Beam Search | 2.0 | 0.7 | 47.5 | 25.3 |
| | DVTS | 2.0 | 0.7 | 49.5 | 26.2 |
| | REBASE | 4.7 | 0.0 | **59.0** | 27.5 |
| | DORA | **6.7** | **2.0** | **59.0** | **31.1** |
| Qwen2.5-1.5B-Instruct | Temperature Sampling | 4.0 | 0.0 | 42.0 | 16.4 |
| | Beam Search | 4.7 | 1.3 | 53.0 | 31.1 |
| | DVTS | 4.7 | 2.0 | 51.5 | 24.9 |
| | REBASE | 5.3 | 2.7 | 54.0 | 32.0 |
| | DORA | **6.7** | **4.0** | **55.0** | **34.2** |

by the performance gains. To this end, we follow Snell et al. (2024), comparing the total FLOPs and inference latency of each method, accounting for the computational cost of the policy model, PRM, and embedding model. Table 2 reports both metrics alongside each method's accuracy. The results demonstrate that DORA is substantially more efficient than all baselines. Specifically, compared to the strongest baseline, REBASE at 256 rollouts, DORA achieves higher accuracy using only 64 rollouts, with a $3.5\times$ reduction in total FLOPs and a $4\times$ speedup in inference latency. These findings suggest that DORA achieves stronger performance with substantially less compute, demonstrating its effectiveness as the most efficient test-time search method.

**DORA enhances search guidance through semantic clustering.** To better understand the effect of DORA's semantic grouping mechanism, we conducted an additional analysis focusing on how well each method improves the intermediate success rate along the reasoning trajectory. Specifically, after each method allocates $K$ reasoning steps, we remove the search algorithm and resume temperature sampling based on the intermediate solutions obtained thus far. We then measure the pass rate of these partial trajectories to estimate the success rate at step $K$. This reflects the method's ability

Table 2: Comparison of FLOPs and inference latency (s) of different methods on MATH500 and AIME24 using LLaMA-3.2-1B-Instruct. All results are reported as mean (standard deviation) over three runs. The best performance for each metric is highlighted in **bold**. Temperature Sampling is excluded due to its significantly lower accuracy.

| Dataset | Method | Rollout | FLOPs | | | | Latency (s) | Accuracy |
| | | | Policy Model | PRM | Embedding Model | Total | | |
| --- | --- | --- | --- | --- | --- | --- | --- | --- |
| MATH500 | Beam Search | 256 | $3.58 \times 10^{14}$ | $2.50 \times 10^{15}$ | 0 | $2.86(0.03) \times 10^{15}$ | 345(7) | 63.6(0.8) |
| | DVTS | 256 | $3.79 \times 10^{14}$ | $2.65 \times 10^{15}$ | 0 | $3.03(0.03) \times 10^{15}$ | 253(8) | 62.0(0.9) |
| | REBASE | 256 | $3.88 \times 10^{14}$ | $2.72 \times 10^{15}$ | 0 | $3.11(0.03) \times 10^{15}$ | 490(10) | 67.4(0.8) |
| | DORA | 64 | $8.45 \times 10^{13}$ | $5.92 \times 10^{14}$ | $2.16 \times 10^{14}$ | $\mathbf{8.92(0.05) \times 10^{14}}$ | **124(8)** | **68.7(0.8)** |
| AIME24 | Beam Search | 256 | $6.83 \times 10^{14}$ | $4.99 \times 10^{15}$ | 0 | $5.67(0.17) \times 10^{15}$ | 816(16) | 11.3(2.8) |
| | DVTS | 256 | $4.74 \times 10^{14}$ | $3.52 \times 10^{15}$ | 0 | $3.99(0.05) \times 10^{15}$ | 734(9) | 11.6(2.4) |
| | REBASE | 256 | $6.61 \times 10^{14}$ | $5.01 \times 10^{15}$ | 0 | $5.67(0.05) \times 10^{15}$ | 978(14) | 14.7(2.3) |
| | DORA | 64 | $4.51 \times 10^{14}$ | $9.86 \times 10^{14}$ | $2.82 \times 10^{14}$ | $\mathbf{1.72(0.19) \times 10^{15}}$ | **240(10)** | **14.7(2.3)** |

Table 3: Intermediate success rate (%) along the reasoning trajectory on MATH500 with $N{=}64$ rollouts, using LLaMA-3.2-1B-Instruct and Qwen2.5-Math-PRM-7B. All results are averaged over 5 runs.

| Step | 0 | 5 | 10 | 15 | 20 | 25 | 30 | 35 | 40 |
| --- | --- | --- | --- | --- | --- | --- | --- | --- | --- |
| Beam Search | 27.7 | 39.3 | 49.6 | 54.2 | 55.8 | 56.8 | 57.3 | 57.5 | 57.5 |
| DVTS | 27.7 | 36.5 | 40.8 | 41.7 | 41.9 | 41.9 | 41.9 | 41.9 | 42.1 |
| REBASE | 27.7 | 39.5 | 51.2 | 54.5 | 56.5 | 57.1 | 58.0 | 58.3 | 58.3 |
| DORA | 27.7 | **40.2** | **51.6** | **55.6** | **57.4** | **58.2** | **59.0** | **59.5** | **59.6** |

to guide the policy model toward more promising reasoning directions early on. As shown in Table 3, DORA consistently achieves the highest pass rates across intermediate steps compared to all baselines. Notably, Step 0 corresponds to the Temperature Sampling baseline (i.e., without any search intervention). Comparing this with the improvements achieved by REBASE and DORA highlights the value of clustering: while both methods significantly outperform the baseline, DORA consistently maintains a lead, suggesting that its semantic clustering mechanism not only reduces redundancy but also enhances the effectiveness of search guidance. We will include this result and analysis in the revised version.

## 5   Related Work

**LLM Test-Time Scaling.** Scaling LLM test-time compute is an effective way to improve performance (OpenAI, 2024). Prior work has explored various strategies, including sampling-based methods with majority voting (Wang et al., 2023) and search-based techniques (Xie et al., 2023; Khanov et al., 2024; Wan et al., 2024). More recently, search algorithms such as breadth-first and depth-first search (Yao et al., 2023), and Monte Carlo Tree Search (MCTS) (Ma et al., 2023; Li et al., 2022; Liu et al., 2023; Choi et al., 2023) have been applied to enhance reasoning. While these methods show promise, many rely on multi-step lookahead operations that are computationally expensive and limit practical scalability (Snell et al., 2024). To improve efficiency, several studies have proposed parallel search strategies (Snell et al., 2024; Beeching et al., 2024; Wu et al., 2025a). Some complementary directions consider branch-and-prune strategies or dynamic decomposition at inference time (Qiu et al., 2024; Light et al., 2025; Li et al., 2024; Wang et al., 2025). However, how to allocate a fixed rollout budget most effectively during search remains underexplored.

**Process Reward Models.** Process reward models (PRMs) have emerged as a powerful tool for improving the reasoning and problem-solving capabilities of large language models. By assigning rewards to intermediate steps, PRMs enable finer-grained evaluation and more effective guidance for multi-step reasoning. They have been shown effective in selecting low-error reasoning traces and providing reward signals for reinforcement-style optimization (Uesato et al., 2022; Polu and Sutskever, 2020; Gudibande et al., 2023). With their rapid development, benchmarks such as ProcessBench (Zheng et al., 2024) and PRMBench (Song et al., 2025) have been introduced to provide comprehensive evaluation protocols. Zhang et al. (2025) further offer practical guidelines for training and deploying PRMs, releasing some of the strongest open-source PRMs to date, particularly for mathematical reasoning.

**Mathematical Reasoning with LLMs.** Recent advances have significantly improved LLMs' performance on mathematical tasks, driven by both training-time and test-time techniques. Training-time methods include large-scale pretraining (OpenAI, 2023; Azerbayev et al., 2024; Shao et al., 2024), supervised fine-tuning (Luo et al., 2023; Tang et al., 2024), and self-improvement via self-generated solutions (Zelikman et al., 2022; Gulcehre et al., 2023; Setlur et al., 2024). Test-time approaches leverage CoT prompting (Wei et al., 2022; Zhao et al., 2025), external tools (Gao et al., 2023; Chen et al., 2023), and self-verification (Weng et al., 2023) to enhance reasoning without changing model weights.

# 6 Conclusions

In this work, we formulate test-time search as a resource allocation problem and derive its optimal solution under a Bayesian framework. Our theoretical analysis offers a unified perspective that explains existing search methods as approximations under varying reward confidence. Furthermore, we find that solution-level allocation favors directions with more candidates and results in suboptimal use of test-time compute. To address this, we propose DORA, a direction-oriented allocation strategy that provably achieves optimality. Extensive experiments on three mathematical reasoning benchmarks demonstrate that DORA consistently improves performance while reducing compute cost. It achieves $3.5\times$ fewer FLOPs and $4\times$ lower latency compared to the strongest baseline REBASE. These results highlight DORA's ability to enhance both the effectiveness and efficiency of test-time inference.

**Limitations.** While our study focuses on scenarios where a process reward model (PRM) is available to evaluate partial trajectories, the underlying framework is not inherently tied to this specific signal. In principle, DORA can incorporate alternative forms of intermediate feedback, such as model confidence or likelihood-based heuristics, extending its applicability beyond PRM-supervised domains. Another limitation is that our theoretical analysis assumes a low-confidence setting, which may not fully capture the dynamics of confidence accumulation during multi-step reasoning. Adapting the allocation strategy to account for increasing confidence over time presents a promising direction for future work.

# Acknowledgements

This work is supported by Beijing Natural Science Foundation (No.4222037, L181010).

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

# A  Details of Parallel Search Process

We present the detailed procedure of the Parallel Search Process in Algorithm 1.

---

**Algorithm 1** Parallel Search Process

---

**Require:** Input problem $x \sim X$, parameters $(\pi, Q, O, V, N)$, step limit $T_{\max}$
**Ensure:** Final Answer
1: $A_0 \leftarrow \{\tau_j = x\}_{j=1}^{N}$                       ▷ Initial active partial solutions
2: $T_{\text{final}} \leftarrow \emptyset$                          ▷ Collected complete solutions
3: **for** $i = 0$ to $T_{\max} - 1$ **do**
4:    **for all** $\tau_j \in A_i$ **do**
5:       Sample action $a \sim \pi(\cdot \mid \tau_j)$
6:       $\tau_j \leftarrow \tau_j \circ a$
7:    **end for**
8:    $T_{\text{final}} \leftarrow T_{\text{final}} \cup \{\tau_j \in A_i \mid \texttt{<EOS>} \in \tau_j\}$       ▷ Add completed solutions
9:    $A_i \leftarrow A_i \setminus \{\tau_j \in A_i \mid \texttt{<EOS>} \in \tau_j\}$       ▷ Remove completed solutions
10:   **if** $|T_{\text{final}}| \geq N$ **then**
11:      **break**
12:   **end if**
13:   Compute PRM scores: $R_j \leftarrow Q(\tau_j)$ for each $\tau_j \in A_i$
14:   Compute rollout allocation: $B_j \leftarrow O(\boldsymbol{R})_j$, where $\boldsymbol{R} = \{R_1, \ldots, R_{|A_i|}\}$
15:   $A_{i+1} \leftarrow \emptyset$
16:   **for** $j = 1$ to $|A_i|$ **do**
17:      Add $B_j$ copies of $\tau_j$ to $A_{i+1}$
18:   **end for**
19: **end for**
20: **return** $V(T_{\text{final}})$                       ▷ Select final answer from complete solutions

---

# B  Proof Section

## B.1  Proof of Proposition 1

Let $p_i \sim \text{Beta}(\kappa w_i, \kappa(1 - w_i))$, where $w_i \in (0, 1)$ is the normalized PRM score for candidate $\tau_i$. Allocating $B_i$ rollouts to candidate $i$, the expected failure probability is

$$\mathbb{E}\left[\prod_{i=1}^{k}(1 - p_i)^{B_i}\right] = \prod_{i=1}^{k} \mathbb{E}\left[(1 - p_i)^{B_i}\right].$$

Using the identity for Beta-distributed $p_i$, we have:

$$\mathbb{E}\left[(1 - p_i)^{B_i}\right] = \prod_{r=0}^{B_i - 1} \frac{\kappa(1 - w_i) + r}{\kappa + r}.$$

Taking the negative logarithm of the success probability, the equivalent optimization problem becomes:

$$\min_{\sum B_i = N} \sum_{i=1}^{k} \sum_{r=0}^{B_i - 1} -\log\left(1 - \frac{\kappa w_i}{\kappa + r}\right).$$

Using the identity $\sum_{r=0}^{n-1} \frac{1}{\kappa + r} = \psi(\kappa + n) - \psi(\kappa)$, where $\psi$ is the digamma function, the objective simplifies to:

$$L(\boldsymbol{B}) = \sum_{i=1}^{k} \kappa w_i \left[\psi(\kappa + B_i) - \psi(\kappa)\right].$$

Relaxing $B_i \in \mathbb{N}$ to $B_i \in \mathbb{R}_{\geq 0}$, we apply the method of Lagrange multipliers with constraint $\sum_i B_i = N$. The partial derivatives yield:

$$\frac{\partial L}{\partial B_i} = \kappa w_i \cdot \psi'(\kappa + B_i),$$

where $\psi'$ is the trigamma function. The KKT condition implies that at optimality:

$$\kappa w_i \cdot \psi'(\kappa + B_i) = \lambda, \quad \text{for all } i \text{ with } B_i > 0, \quad \text{and} \quad \sum B_i = N.$$

We now analyze three asymptotic regimes of $\kappa$:

**Case 1: Fixed finite** $\kappa > 0$ Using the approximation $\psi'(\kappa + B_i) \approx \frac{1}{\kappa + B_i}$ when $\kappa + B_i \gg 1$, the optimality condition becomes:

$$\frac{\kappa w_i}{\kappa + B_i} \approx \lambda \quad \Rightarrow \quad B_i^\star \approx \frac{\kappa w_i}{\lambda} - \kappa.$$

Summing both sides over $i$ and enforcing $\sum_i B_i = N$, we solve for $\lambda \approx \kappa/(N + k\kappa)$, yielding:

$$\boxed{B_i^\star \approx (N + k\kappa)w_i - \kappa.}$$

**Case 2:** $\kappa \to \infty$ In this regime, the Beta prior becomes increasingly concentrated at $p_i = w_i$. Hence,

$$\mathbb{E}[(1 - p_i)^{B_i}] \to (1 - w_i)^{B_i}, \quad \text{and} \quad \mathbb{E}\left[\prod_{i=1}^{k}(1 - p_i)^{B_i}\right] \to \prod_{i=1}^{k}(1 - w_i)^{B_i}.$$

To minimize failure probability, we solve:

$$\min_{\sum B_i = N} \sum_{i=1}^{k} B_i \log(1 - w_i).$$

Since $\log(1 - w_i) < 0$, this is minimized by allocating all rollouts to the candidate with the largest $w_i$, i.e.,

$$O^\star(\boldsymbol{w})_i = \begin{cases} N, & \text{if } i = \arg\max_j w_j, \\ 0, & \text{otherwise.} \end{cases}$$

**Case 3:** $\kappa \to 0$ In this regime, the Beta distribution becomes highly uncertain:

$$p_i \sim \text{Beta}(\kappa w_i, \kappa(1 - w_i)) \xrightarrow[\kappa \to 0]{} \begin{cases} 1, & \text{with probability } w_i, \\ 0, & \text{with probability } 1 - w_i. \end{cases}$$

Hence,

$$\mathbb{E}\left[(1 - p_i)^{B_i}\right] \to \begin{cases} 1 - w_i, & \text{if } B_i > 0, \\ 1, & \text{if } B_i = 0. \end{cases}$$

Thus, the expected failure probability becomes:

$$\prod_{i=1}^{k} \mathbb{E}\left[(1 - p_i)^{B_i}\right] \to \prod_{i:B_i > 0}(1 - w_i),$$

which depends only on whether a candidate receives at least one rollout, not how many. To minimize failure, we must select a subset $S \subseteq \{1, \ldots, k\}$ with $|S| \le N$ such that:

$$\prod_{i \in S}(1 - w_i)$$

is minimized. This is achieved by choosing the top-$s = \min(k, N)$ candidates with the largest $w_i$. Then the optimal allocation is:

$$O^\star(\boldsymbol{w})_i = \begin{cases} 1, & \text{if } i \in \text{Top-}s \text{ of } w_i, \\ 0, & \text{otherwise,} \end{cases} \quad \text{with remaining } N - s \text{ rollouts arbitrarily assigned.}$$

## B.2 Proof of Proposition 2

In the $\kappa \to 0$ regime, Proposition 1 shows that the expected success probability is maximized by the solution:

$$B_i \propto w_i, \quad \text{where } w_i = \frac{e^{R_i}}{\sum_j e^{R_j}}.$$

This corresponds to maximizing the log-utility objective:

$$\mathcal{L} = \sum_{i=1}^{k} w_i \log B_i.$$

To analyze the effect of structural redundancy, we group candidate solutions into $g$ reasoning directions. Let direction $j$ contain $k_j$ candidates, each with identical score $R_j$, and index set $\mathcal{E}_j$.

The optimal direction-aware allocation follows:

$$Q_j := \frac{e^{R_j}}{\sum_{l=1}^{g} e^{R_l}}, \qquad B_j^{(\text{direction})} := N \cdot Q_j.$$

The corresponding log-utility is:

$$\mathcal{L}^{(\text{dir})} = \sum_{j=1}^{g} Q_j \log B_j^{(\text{direction})} = \log N + \sum_{j=1}^{g} Q_j \log Q_j.$$

REBASE assigns each candidate $i \in \mathcal{E}_j$ rollout weight:

$$w_i = \frac{e^{R_j}}{\sum_{l=1}^{g} k_l e^{R_l}}, \quad \text{so} \quad B_j^{(\text{solution})} = \sum_{i \in \mathcal{E}_j} N w_i = N \cdot \frac{k_j e^{R_j}}{\sum_{l=1}^{g} k_l e^{R_l}}.$$

This induces a direction-level distribution:

$$\hat{Q}_j := \frac{k_j e^{R_j}}{\sum_{l=1}^{g} k_l e^{R_l}}.$$

The resulting utility is:

$$\mathcal{L}^{(\text{sol})} = \sum_{j=1}^{g} Q_j \log B_j^{(\text{solution})} = \log N + \sum_{j=1}^{g} Q_j \log \hat{Q}_j.$$

The gap in log-utility is:

$$\mathcal{L}^{(\text{dir})} - \mathcal{L}^{(\text{sol})} = \sum_{j=1}^{g} Q_j \log \frac{Q_j}{\hat{Q}_j} = \text{KL}(Q \parallel \hat{Q}) \geq 0.$$

Equality holds if and only if $Q_j = \hat{Q}_j$ for all $j$, i.e.,

$$\frac{e^{R_j}}{\sum_l e^{R_l}} = \frac{k_j e^{R_j}}{\sum_l k_l e^{R_l}} \quad \Rightarrow \quad k_j = k \text{ for all } j.$$

Thus, the solution-level allocation is suboptimal unless all reasoning directions contain the same number of candidate solutions.

## B.3 Proof of Theorem 1

Assume candidate solutions are partitioned into $g$ reasoning directions, where direction $j \in \{1, \ldots, g\}$ contains $k_j$ candidates indexed by $\mathcal{E}_j$, and all candidates in $\mathcal{E}_j$ share the same PRM score $R_j$.

Under REBASE, softmax is computed at the solution level:

$$\tilde{q}_i = \frac{e^{R_j}}{\sum_{l=1}^{g} k_l e^{R_l}}, \quad \text{for } i \in \mathcal{E}_j.$$

Aggregating across each direction yields the induced direction-level distribution:

$$\hat{Q}_j^{\text{REBASE}} = \sum_{i \in \mathcal{E}_j} \tilde{q}_i = \frac{k_j e^{R_j}}{\sum_{l=1}^{g} k_l e^{R_l}}.$$

To eliminate the bias from uneven candidate counts $k_j$, DORA reweights each $\tilde{q}_i$ by the inverse of its cluster size:

$$\hat{q}_i = \frac{\tilde{q}_i}{k_j}, \quad \text{for } i \in \mathcal{E}_j.$$

The normalization constant becomes:

$$Z = \sum_{i=1}^{k} \hat{q}_i = \sum_{j=1}^{g} \sum_{i \in \mathcal{E}_j} \frac{\tilde{q}_i}{k_j} = \sum_{j=1}^{g} \frac{k_j e^{R_j}}{k_j \sum_{l=1}^{g} k_l e^{R_l}} = \frac{\sum_{j=1}^{g} e^{R_j}}{\sum_{l=1}^{g} k_l e^{R_l}}.$$

Normalizing gives the final corrected weight:

$$\hat{q}_i^{\text{final}} = \frac{\hat{q}_i}{Z} = \frac{e^{R_j}}{k_j \sum_{l=1}^{g} e^{R_l}}, \quad \text{for } i \in \mathcal{E}_j.$$

Aggregating over direction $j$, the direction-level allocation becomes:

$$\hat{Q}_j^{\text{final}} = \sum_{i \in \mathcal{E}_j} \hat{q}_i^{\text{final}} = k_j \cdot \frac{e^{R_j}}{k_j \sum_{l=1}^{g} e^{R_l}} = \frac{e^{R_j}}{\sum_{l=1}^{g} e^{R_l}} = Q_j.$$

Thus, the final allocation satisfies

$$\sum_{i \in \mathcal{E}_j} B_i \propto Q_j,$$

which exactly matches the optimal direction-level allocation given in Eq. 9.

## C  Details of Beta Distribution

The Beta distribution is a standard choice for modeling random variables on the unit interval, and its parameters $(\alpha, \beta) = (\kappa w_i, \kappa(1 - w_i))$ are interpretable: the mean is $\mathbb{E}[p_i] = w_i$, and the variance is inversely related to $\kappa$. Specifically:

- When $\kappa$ is small, the distribution is diffuse and uncertain.
- When $\kappa$ is large, the distribution is sharply peaked around $w_i$, indicating high confidence.

Figure 4 visualizes the effect of different $\kappa$ values with $w_i$ fixed at 0.7.

## D  More Experiments

### D.1  DORA provides larger gains on harder problems

Figure 5 shows that while DORA remains the top-performing strategy across the entire MATH500 benchmark, the size of its advantage depends sharply on difficulty. On easier Level 1–2 problems, most methods perform well given moderate rollout budgets, so the accuracy curves for all methods converge closely. On the other hand, on harder Level 3–5 problems, the gap between DORA and solution-level methods widens steadily with budget, with DORA achieving a clear lead at higher rollout levels. We hypothesize that harder problems amplify DORA's strength as they typically require longer reasoning chains (Wu et al., 2025b), which allows more opportunities for rollout allocation across search steps. As the number of allocation rounds increases, a principled strategy like DORA could compound its advantage by continually prioritizing promising directions and avoiding wasted computation.

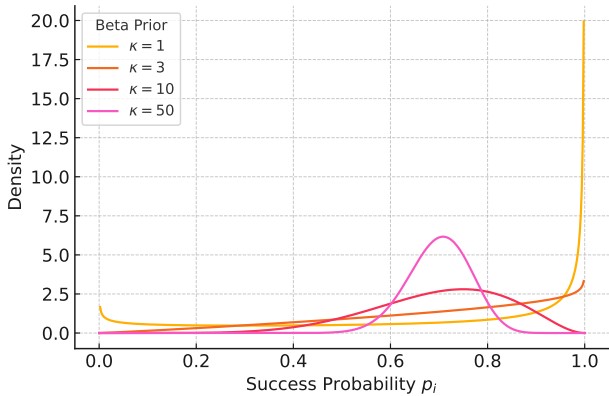

Figure 4: Effect of the concentration parameter $\kappa$ on the Beta prior. All curves are plotted with fixed mean $w_i = 0.7$. Larger $\kappa$ yields a more concentrated prior around $w_i$, while smaller $\kappa$ reflects greater uncertainty.

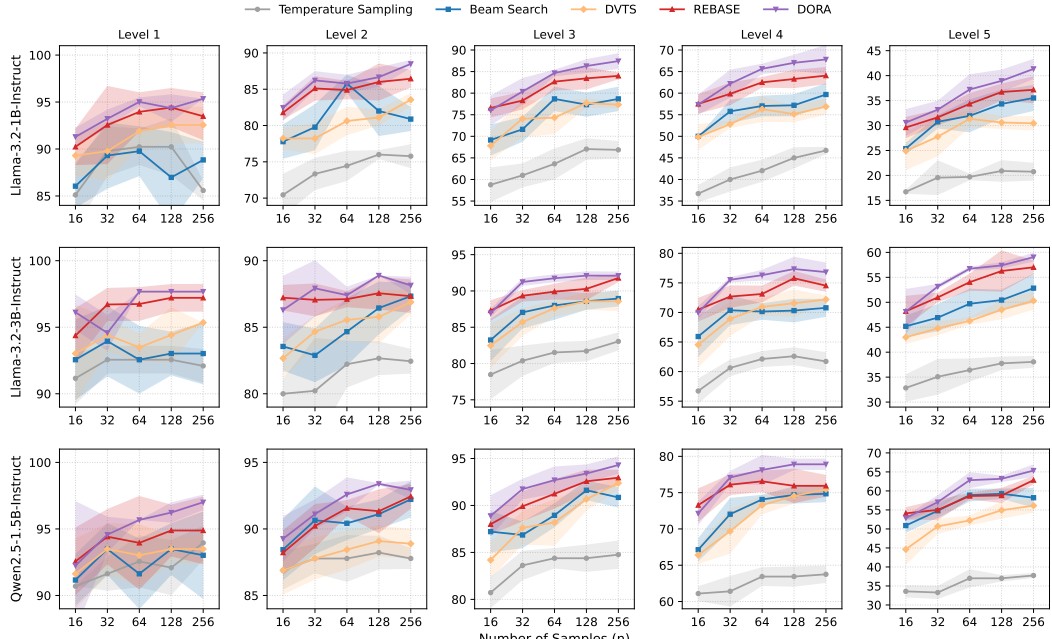

Figure 5: Comparison of method accuracy on MATH500 across different difficulty levels.

## D.2 Ablation on Clustering Design

To better isolate the effect of DORA's semantic clustering mechanism, we conduct an ablation study comparing several alternative clustering strategies before finalizing the soft clustering design.

**KMeans clustering.** KMeans requires predefining the number of clusters $K$. However, the actual number of distinct reasoning directions can vary substantially across problems and reasoning steps, making a fixed $K$ a poor approximation and often leading to suboptimal clustering performance.

**Hierarchical clustering.** We experiment with hierarchical clustering, which avoids setting $K$ by merging solutions based on a fixed cosine similarity threshold $C$. However, the optimal value of $C$ should ideally vary with the depth and complexity of reasoning, which differ significantly across problems. This sensitivity made hierarchical clustering even less effective than KMeans in our experiments.

Table 4: Ablation study of different clustering methods on MATH500 with $N{=}64$ rollouts, using LLaMA-3.2-1B-Instruct. Results are averaged over 5 runs.

| Cluster Method | K=3 | K=5 | K=10 | C=0.95 | C=0.90 | C=0.80 | Soft (default) |
|---|---|---|---|---|---|---|---|
| Accuracy | 67.6 | 68.0 | 67.8 | 67.0 | 66.6 | 66.0 | **68.7** |

**Soft clustering (ours).** Based on these findings, we adopted a soft clustering approach in DORA, which avoids hard assignment boundaries and allows for more adaptive and robust semantic grouping across diverse reasoning structures.

As shown in Table 4, the soft clustering variant achieves the best performance, outperforming all hard clustering alternatives. This demonstrates that allowing soft, overlapping group assignments enables DORA to better adapt to the variable structure of reasoning paths, improving robustness and accuracy across diverse problem types.

## D.3 DORA generalizes across model scales and PRM families

As shown in Table 5, we further test DORA with a larger policy–PRM pair: LLaMA-3.1-8B-Instruct guided by Qwen2.5-Math-PRM-72B on MATH500 with $N{=}64$ rollouts. DORA attains the highest accuracy (80.6), outperforming Beam Search (78.2), DVTS (78.4), and REBASE (79.2). We also replace the reward model with a different PRM family (Skywork-PRM-7B) and observe consistent gains across all policy models (Table 6), surpassing the strongest baseline in each case. These results indicate that DORA's advantages persist when scaling to larger models and when switching across distinct reward-model families, underscoring its robustness as a reliable test-time search strategy.

Table 5: Accuracy on MATH500 with a larger policy model and PRM: LLaMA-3.1-8B-Instruct as the policy and Qwen2.5-Math-PRM-72B as the PRM ($N{=}64$ rollouts). Best results are in **bold**.

| PRM | Policy Model | Temperature Sampling | Beam Search | DVTS | REBASE | DORA |
|---|---|---|---|---|---|---|
| Qwen2.5-Math-PRM-72B | LLaMA-3.1-8B-Instruct | 70.5 | 78.2 | 78.4 | 79.2 | **80.6** |

Table 6: Accuracy on MATH500 with an alternative PRM family (Skywork-PRM-7B) at $N{=}64$ rollouts. Best results are in **bold**.

| Method | LLaMA-3.2-1B-Instruct | LLaMA-3.2-3B-Instruct | Qwen2.5-1.5B-Instruct |
|---|---|---|---|
| Temperature Sampling | 58.5 | 69.5 | 73.0 |
| Beam Search | 69.0 | 75.5 | 79.7 |
| DVTS | 66.2 | 74.2 | 78.9 |
| REBASE | 70.8 | 76.2 | 80.2 |
| DORA | **71.8** | **76.6** | **81.0** |

## D.4 DORA is robust to hyperparameter choices

We further study the temperature parameters $T_b$ (softmax over directions) and $T_s$ (semantic similarity) on MATH500 with $N{=}64$ (Table 7). Both REBASE and DORA perform well at $T_b \in \{0.01, 0.1\}$ but degrade at $T_b{=}1.0$, where the softened distribution weakens PRM guidance and approaches unguided sampling. Importantly, DORA is stable across a wide range of $T_s$ values (0.001–1.0), indicating low sensitivity to the clustering threshold. To further investigate DORA's robustness across retriever families, we compared our default retriever (bge-m3, 568M parameters) with two popular alternatives from the MTEB leaderboard (Muennighoff et al., 2023) that support long inputs (2048 tokens): e5-base-4k (110M) and gte-multilingual-base (305M). As shown in Table 8, BGE and GTE deliver comparable performance across all policy models, while E5 is slightly worse—likely due to its smaller capacity and thus weaker clustering in high-dimensional embedding space. Together, these results show that DORA's gains are not brittle: it maintains strong accuracy under reasonable choices of temperatures and retrievers, reinforcing its practicality for real-world deployment.

Table 7: Sensitivity analysis of temperature hyperparameters $T_b$ and $T_s$ on MATH500 with $N{=}64$ rollouts using LLaMA-3.2-1B-Instruct. All results are averaged over 5 runs.

| Method | $T_b$ | | | $T_s$ | | | |
|--------|------|-----|-----|-------|------|-----|-----|
|        | 0.01 | 0.1 | 1.0 | 0.001 | 0.01 | 0.1 | 1.0 |
| REBASE | 64.8 | 65.9 | 55.4 | - | - | - | - |
| DORA   | 67.4 | 68.7 | 57.2 | 67.8 | 68.7 | 68.0 | 67.5 |

Table 8: Ablation on embedding (retriever) models on MATH500 with $N{=}64$ rollouts. We compare our default BGE-M3 to GTE-multilingual-base and E5-base-4k. Results are averaged over 5 runs.

| Policy Model | BGE | GTE | E5 |
|--------------|-----|-----|-----|
| LLaMA-3.2-1B-Instruct | 68.7 | 68.2 | 66.8 |
| LLaMA-3.2-3B-Instruct | 77.6 | 77.4 | 76.8 |
| Qwen-2.5-1.5B-Instruct | 80.8 | 80.2 | 79.2 |

# E Implementation Details

## E.1 Experimental Hyperparameters

All experiments use temperature sampling with `temperature = 0.8` and `top_p = 1.0`. We set the token limit to 256 per step and 2048 tokens in total for each solution. For Beam Search and DVTS, we use a beam width of 4 following Snell et al. (2024). For REBASE, we set its $T_b$ to 0.1, consistent with its original implementation. For DORA, we employ the open-source BGE-M3 embedding model (Chen et al., 2024a) to compute semantic similarity between trajectories, chosen for its lightweight architecture, strong empirical performance, and ability to handle long input sequences. We set the $T_b$ for quality scores to 0.1 (matching REBASE), and the semantic similarity temperature $T_s$ to 0.01. All experiments are executed in parallel on a cluster with 32 NVIDIA A100 GPUs (40G), where each individual run is allocated to a single GPU.

## E.2 Details of Prompt

Following Beeching et al. (2024), we employ the prompt below for LLM mathematical reasoning:

```
Solve the following math problem efficiently and clearly:

- For simple problems (two steps or fewer):
  Provide a concise solution with minimal explanation.

- For complex problems (three steps or more):
  Use this step-by-step format:

## Step 1: [Concise description]
[Brief explanation and calculations]

...
## Step 2: ...

Regardless of problem complexity, always conclude with:
Therefore, the final answer is: \boxed{answer}.
```

