# OpenReview forum: "Every Rollout Counts: Optimal Resource Allocation for Efficient Test-Time Scaling"
_NeurIPS.cc/2025/Conference — NeurIPS 2025 poster_

### Official Review · Reviewer_T3Po · 2025-06-05

**Clarity:** 3
**Significance:** 2
**Originality:** 3
**Rating:** 5
**Confidence:** 3

**Summary:**

In this paper, the authors address a fundamental  aspect of TTS, specifically the optimal allocation of computational resources (rollouts) to maximise the success probability for LLM during inference. Existing approaches, including beam search, DVTS, and REBASE, tend to allocate resources at the solution level, unintentionally favouring reasoning directions with more candidate solutions. To rectify this inefficiency, the authors propose DORA, a method that allocates resources based on distinct reasoning directions rather than individual candidate solutions. DORA leverages semantic embeddings and a soft clustering approach to identify and balance resources across reasoning directions, theoretically proven optimal under certain assumptions. Extensive experiments conducted on challenging mathematical reasoning tasks (MATH500, AIME2024, AIME2025) demonstrate DORA’s superiority over baseline methods to some extent.

**Questions:**

1. How sensitive is DORA’s performance to the choice of hyperparameters (e.g., temperature parameters)? Is there any guidance or robustness analysis for selecting these parameters?
2. Could you provide insights into the scalability of DORA, particularly how the embedding overhead scales with very large numbers of candidate solutions?

**Ethical Concerns:**

["NO or VERY MINOR ethics concerns only"]

**Final Justification:**

The authors have provided substantial experiments that address my major concerns. Kindly ensure these are reflected in the revision.

**Limitations:**

yes

**Quality:**

3

**Strengths And Weaknesses:**

### **Pros:**

1. The derivation of optimal strategies through Bayesian probabilistic modeling provides solid theoretical backing.
2. Well-written and well-structured paper.

### **Cons:**

1. The claim "DORA is the most effective parallel search method" appears overly strong. The paper evaluates DORA exclusively using a single reward model (Qwen2.5-Math-PRM-7B), three relatively small-scale LLMs (<7B), and on a narrow set of benchmarks limited to mathematical reasoning datasets (MATH500, AIME2024, and AIME2025). Such a restricted evaluation setup raises concerns about the generality and robustness of the proposed method. To strengthen this claim, the authors should include additional experiments using alternative reward models and more diverse datasets.

2. While DORA introduces embedding-based clustering to achieve optimal allocation, the computational overhead of embedding calculations, especially in scenarios involving a large number of candidates, is unclear.

3. The reliance on soft clustering and temperature parameters (e.g., allocation temperature) suggests potential sensitivity to hyperparameter settings, yet detailed sensitivity analysis is missing.

---

> ### Author Rebuttal · Authors · 2025-07-31
>
> Dear Reviewer T3Po, we sincerely appreciate you taking the time to read our manuscript and provide valuable feedback. Below are our responses to the concerns you raised, which will be incorporated into the updated version to further enhance the quality of our submission.
>
> ### **C1: On the Generality of the Evaluation**
> We appreciate your thoughtful feedback regarding the generality and robustness of DORA. We fully agree that evaluating across more diverse PRMs, policy models, and benchmarks is crucial to substantiating the claim of DORA’s effectiveness as a parallel search method.
> To address this, we have conducted a new series of experiments beyond the initial setup, expanding the evaluation in three directions (All results are averaged over 5 independent runs to ensure statistical stability):
>
> (1) Broader Benchmark Coverage: We incorporated four additional standard math reasoning benchmarks: HMMT24, HMMT25, AMC23, and AMC24. Under a rollout budget of N=64, the results are summarized as follows:
>
> |Policy Model|Method|HMMT24|HMMT25|AMC23|AMC24|
> |------------|------|--------|--------|--------|--------|
> |LLaMA-3.2-1B|Temperature Sampling |0.0     |0.0     |28.0    |13.3    |
> ||Beam Search |2.0     |0.0     |43.0    |15.5    |
> ||DVTS  |1.3     |0.0     |41.0    |18.0    |
> ||REBASE|2.0     |0.0     |43.0    |18.2    |
> ||DORA  |**3.3** |0.0 |**45.0**|**20.0**|
> |LLaMA-3.2-3B|Temperature Sampling   |0.7     |0.0     |48.5    |21.7    |
> ||Beam Search  |2.0     |0.7     |55.5    |25.3    |
> ||DVTS  |2.0     |0.0     |57.0    |26.2    |
> ||REBASE|4.7     |0.0     |**59.0**|27.5    |
> ||DORA  |**6.7** |**2.0** |**59.0**|**31.1**|
> |Qwen-2.5-1.5B|Temperature Sampling   |4.0     |0.0     |42.0    |16.4    |
> ||Beam Search  |4.7     |1.3     |53.0    |31.1    |
> ||DVTS  |4.7     |2.0     |51.5    |24.9    |
> ||REBASE|5.3     |2.7     |54.0    |32.0    |
> ||DORA  |**6.7** |**4.0** |**55.0**|**34.2**|
>
>
> (2) Larger Policy Models and PRM: We evaluated DORA with more powerful models by using Qwen2.5-Math-PRM-72B as the PRM to guide the LLaMA-3.1-8B-Instruct policy model on the MATH500 benchmark with N=64. The performance is reported below:
>
> |PRM |Policy Model|Temperature Sampling|Beam Search|DVTS|REBASE|DORA|
> |-----|-----|----|----|-----|------|-----|
> |Qwen2.5-Math-PRM-72B|LLaMA-3.1-8B|70.5|78.2|78.4|79.2|**80.6**|
>
> (3) Different PRM Families: To evaluate generalization across PRM families, we additionally tested **Skywork-PRM-7B** on the MATH500 benchmark with N=64. The results are shown below:
>
> |Method|LLaMA-3.2-1B|LLaMA-3.2-3B|Qwen-2.5-1.5B|
> |------|------------|------------|-------------|
> |Temperature Sampling|58.5|69.5|73.0|
> |Beam Search|69.0|75.5|79.7|
> |DVTS|66.2|74.2|78.9|
> |REBASE|70.8|76.2|80.2|
> |DORA|**71.8**|**76.6**|**81.0**|
>
> Above experiments demonstrate that DORA generalizes well across a broader range of benchmarks, scales to larger policy models and PRMs, and is compatible with PRMs from different families. Specifically, the experiments with Qwen2.5-Math-PRM-72B were executed in parallel on a cluster with 100 H20 GPUs (96GB each), using 2 GPUs per run, and were completed in approximately 4 days. We will include these results and analysis in the later version.
>
> ### **C2 & Q2: About computational overhead of embedding calculations with a large number of candidates**
>
> Thank you for bringing up this important point. We agree that analyzing the computational overhead of embedding is essential, especially when scaling DORA to larger rollout budgets. According to \[1-2], the total FLOPs incurred during Test-Time Scaling (TTS) can be calculated as:
>
> $$\text{Total FLOPs} = 2N \cdot T_{\text{sol}} \cdot \left( P_{\text{policy}} + P_{\text{PRM}} + P_{\text{emb}} \right)$$
>
> where:
>
> * $N$: rollout budget (i.e., number of sampled solutions),
> * $T_{\text{sol}}$: average number of tokens per solution,
> * $P_{\text{policy}}, P_{\text{PRM}}, P_{\text{emb}}$: parameter sizes of the policy model, PRM, and embedding model, respectively.
>
> The relative overhead of the embedding model in the total FLOPs is thus:
>
> $$Ratio = \frac{P_{\text{emb}}}{P_{\text{policy}} + P_{\text{PRM}} + P_{\text{emb}}}$$
>
> This fraction is constant and does **not** grow with $N$, implying that the embedding overhead scales **linearly** just like the other components and remains proportionally small.
>
> In practice, Parameter of embedding models used in our setup (e.g., BGE-M3 <600M) are much smaller than that of the policy model (e.g., Qwen2.5-1.5B-Instruct) and the PRM (e.g., Qwen2.5-Math-PRM-7B).The additional computational burden introduced by the embedding model would not be a major concern as the rollout budget $N$ increases. We will include this analysis in the revised version.
>
>
> ### **C3 & Q1: On the Sensitivity to Temperature Hyperparameters**
>
> We appreciate your concern regarding DORA’s reliance on temperature-based mechanisms and the associated sensitivity to hyperparameter choices. To address this, we conducted a detailed ablation study on two key temperature parameters:
>
> (1) The allocation temperature $T_b$, which controls the sharpness of the softmax when allocating rollout budgets;
>
> (2) The semantic similarity temperature $T_s$, which determines how close two solutions  should be to be considered as belonging to the same reasoning direction.
>
> For $T_b$, we default to 0.1 to align with REBASE, ensuring consistency and fairness. For $T_s$, we selected the final value via empirical study. We acknowledge that these results were not included in the original submission and apologize for the oversight.
>
> We now report the full sensitivity analysis on the MATH500 benchmark with rollout budget N=64, using LLaMA-3.1-8B-Instruct as the policy model. Each configuration was run 5 times with different seeds, and we report the averaged accuracy:
>
> |$T_b$   |0.01|0.1|1.0|
> |------|----|----|----|
> |REBASE|64.8|65.9|55.4|
> |DORA  |67.4|68.7|57.2|
>
> |$T_s$  |0.001|0.01|0.1|1.0|
> |-----|-----|----|----|----|
> |DORA|67.8 |68.7|68.0|67.5|
>
> These results suggest that DORA is relatively robust to the choice of $T_s$, with accuracy remaining stable across a wide range. In contrast, both DORA and REBASE show more noticeable degradation when $T_b=1.0$, as the softmax becomes too flat to leverage PRM guidance effectively. In the extreme case where $T_b \to \infty$, the model degenerates into temperature sampling. We will include these results and analysis in the updated version to provide clearer guidance for future applications.
>
> [1] Scaling Laws for Neural Language Models, OpenAI.
>
> [2] Scaling LLM Test-Time Compute Optimally Can be More Effective than Scaling Parameters for Reasoning, ICLR2025.

---

> ### Author Response · Authors · 2025-08-06
>
> Dear Reviewer T3Po,
>
> We hope this message finds you well. As the discussion period is nearing its end, we would like to kindly follow up to ensure that our response has addressed your concerns satisfactorily. If there are any additional questions or suggestions you would like us to consider, please do not hesitate to let us know. Your insights are invaluable to us, and we would be more than happy to engage in further discussion to improve our work.
>
> Thank you once again for your time and thoughtful feedback！
>
> Best regards, The Authors

---

> ### Author Response · Authors · 2025-08-08
> **Gentle Reminder**
>
> Dear Reviewer T3Po,
>
> We are writing to you with a gentle follow-up as the discussion period will close in about 30 hours. We completely understand that your time is precious and you are likely very busy. We just wanted to kindly reiterate that if you have any final comments or concerns, we are fully prepared to do whatever is necessary to address them promptly.
>
> Thank you once again for your time and invaluable contribution to our work.
>
> Best regards,
>
> Authors

---

> > ### Comment · Reviewer_T3Po · 2025-08-08
> > **Thanks for the response**
> >
> > Many thanks for your detailed responses and additional experiments on more benchmarks and larger models. I am happy to increase my score.

---

> > > ### Author Response · Authors · 2025-08-08
> > > **Reply to Reviewer T3Po**
> > >
> > > Thank you for your prompt reply and your recognition of our work! We will make sure to incorporate the above discussion into the final version of our paper. Once again, we sincerely appreciate the time and effort you dedicated to reviewing our submission. Your insights are truly invaluable, and we are confident they will significantly enhance the quality of our work.

---

### Official Review · Reviewer_JSmU · 2025-07-02

**Clarity:** 3
**Significance:** 2
**Originality:** 3
**Rating:** 4
**Confidence:** 3

**Summary:**

This paper investigates the problem of test-time rollout allocation for large language models (LLMs) under a fixed computational budget. The authors formalize the rollout allocation process as a resource optimization problem, and propose a novel method called DORA. Experiments on challenging mathematical benchmarks show consistent performance gains over existing methods.

**Questions:**

(1)	Assumption of soft clustering quality: DORA relies on semantic embeddings and soft clustering via cosine similarity. However, it is unclear how robust this semantic grouping is across tasks and models. How sensitive is DORA to the choice of embedding model and the temperature parameters $T_s$?

(2)	Scalability concerns: While the reported FLOPs and latency gains are encouraging, DORA introduces an extra embedding + affinity matrix computation step. How does this overhead scale with very large numbers of candidate trajectories (e.g., >10,000) which may lead to a failure case?

(3)	Assumption of shared PRM score per direction: The theoretical justification of DORA assumes that all solutions in a direction $j$ share the same PRM score $R_j$. In practice, PRM scores are noisy and not strictly aligned. How is this mismatch handled, and what is the empirical impact when intra-directional variance in PRM scores is high?

(4)	No ablation on clustering effect: The paper does not include ablation studies comparing with other clustering methods at all. This would help isolate whether the gains come from semantic direction-aware allocation, or merely from reduced redundancy.

(5)	Limited discussion of generalization: The method is evaluated only on mathematical reasoning tasks. While these are valid and difficult benchmarks, it remains unclear whether DORA generalizes to other domains like code generation, etc.

**Ethical Concerns:**

["NO or VERY MINOR ethics concerns only"]

**Limitations:**

While this paper tackles an important problem with an elegant solution and promising results, several key assumptions require further clarification and justification.

**Quality:**

3

**Strengths And Weaknesses:**

The paper tackles a practically important yet underexplored problem, the proposed direction-aware allocation strategy is novel and effectively addresses a real shortcoming in rollout allocation, showing state-of-the-art performance on multiple mathematical reasoning tasks and demonstrates superior compute efficiency.

---

> ### Author Rebuttal · Authors · 2025-07-31
>
> Dear Reviewer JSmU, we sincerely appreciate you taking the time to read our manuscript and provide valuable feedback. Below are our responses to the concerns you raised, which will be incorporated into the updated version to further enhance the quality of our submission.
>
> ### **Q1: On the Sensitivity to the Embedding Model and Semantic Similarity Temperature $T_s$**
>
> Thank you for this important question. We fully understand your concern regarding the robustness of DORA’s semantic grouping, which relies on the embedding model and the soft clustering temperature $T_s$.
>
> To evaluate the impact of different retriever (embedding) models, we compared our original setup (BGE-M3, 568M parameters) with two widely used alternatives from the MTEB leaderboard [1] that support long inputs (≥2048 tokens): E5-base-4k (110M) and GTE-multilingual-base (305M). All experiments were conducted on MATH500 with $N=64$, and all other settings were kept identical to those in the main experiments. Results were averaged over 5 independent runs:
>
> |Policy Model|BGE|GTE|E5|
> |-------------|----|----|----|
> |LLaMA-3.2-1B|68.7|68.2|66.8|
> |LLaMA-3.2-3B|77.6|77.4|76.8|
> |Qwen-2.5-1.5B|80.8|80.2|79.2|
>
> The results show that GTE and BGE yield comparable performance, indicating that DORA is robust to retriever changes within similar capacity ranges. However, E5 exhibits certain performance degradation, which we hypothesize is due to its smaller model size and thus weaker clustering capability in high-dimensional embedding space.
>
> We further conducted an ablation study to evaluate the sensitivity of DORA to the semantic similarity temperature $T_s$, which determines how close two solutions should be to be considered as belonging to the same reasoning direction. Experiments were run on MATH500 with $N=64$, using LLaMA-3.1-8B-Instruct as the policy model. The results (averaged over 5 runs) are shown below:
>
> |$T_s$|0.001|0.01|0.1|1.0|
> |-----|-----|----|----|----|
> |DORA|67.8|68.7|68.0|67.5|
>
> These findings demonstrate that DORA is relatively robust to the choice of $T_s$. We will include these results and corresponding discussion in the updated version to better clarify DORA’s stability with respect to embedding models and soft clustering settings.
>
> ### **Q2: On the Scalability of Embedding and Affinity Computation with Very Large Candidate Sets**
>
> Thank you for raising this important point. We agree that it is essential to assess the overhead introduced by embedding and affinity matrix computation, particularly at large rollout scales.
>
> Following \[2–3], the total FLOPs of Test-Time Scaling (TTS) can be estimated as:
>
> $$\text{Total FLOPs} = 2N \cdot T_{\text{sol}} \cdot (P_{\text{policy}} + P_{\text{PRM}} + P_{\text{emb}})$$
>
> where $N$ is the rollout budget, $T_{\text{sol}}$ is the average solution length, and $P_{\text{policy}}, P_{\text{PRM}}, P_{\text{emb}}$ denote parameter counts of the policy model, PRM, and embedding model respectively.
>
> The embedding overhead ratio in TTS is caculated by:
>
> $$\text{Ratio} = \frac{P_{\text{emb}}}{P_{\text{policy}} + P_{\text{PRM}} + P_{\text{emb}}}$$
>
> This ratio is **independent of $N$**, indicating that embedding cost scales linearly with rollout size and remains proportionally small. In our setup, the embedding model (e.g., BGE-M3 <600M) is significantly smaller than the policy (e.g., Qwen2.5-1.5B) and PRM (e.g., Qwen2.5-Math-PRM-7B), so its contribution to total cost remains minor.
>
> We also analyzed **memory and runtime cost** of computing the affinity matrix (i.e., pairwise cosine similarities). When $N = 10{,}000$ (a scale well beyond most prior TTS works, which typically use $N \leq 2048$), with 1024-dim embeddings in fp16:
>
> * Matrix size: $10^8$ entries → 200 MB
> * Embedding storage: 20 MB
> * Total memory: **\~220 MB**, easily handled by modern CPUs
>
> For runtime, computing 100M cosine similarities (each a 1024-dim dot product) totals \~200 GFLOPs. On a single CPU core this takes \~6–7 seconds, or <0.5s with 16-core parallelization. Since this is a **one-time cost per rollout**, it does not meaningfully affect overall latency.
>
> We will include this analysis in the final version to clarify that DORA’s embedding-related steps scale efficiently even with large candidate sets.
>
> ### **Q3: On the Assumption of Shared PRM Scores Within a Direction**
>
> Thank you for this thoughtful question. The assumption in Theorem 1—that all solutions within a semantic direction are perfectly correlated and share the same PRM score—is introduced solely to enable a tractable theoretical analysis，which allows us to derive a clean optimality result.
>
> In practice, solutions grouped within a direction are typically only partially correlated, and their PRM scores can differ due to modeling noise and semantic drift. Explicitly modeling this intra-directional variance would render the analysis intractable. To address this, DORA adopts a soft clustering mechanism and retains per-candidate PRM scores $R_i$, without collapsing them to a single cluster-level score.
>
> Rollout allocation is computed individually for each solution using the formula (lines 195–196):
>
> $$w_i' = \frac{w_i \cdot \gamma_i}{\sum_j w_j \cdot \gamma_j},$$
>
> where $w_i$ reflects solution quality, and $\gamma_i$ captures semantic uniqueness from the affinity matrix. This ensures that higher-quality candidates within a direction can still receive more rollouts, even when grouped with lower-scoring neighbors. By combining soft clustering with individualized scoring, DORA remains robust to intra-directional PRM variance. Our empirical results across multiple benchmarks show that DORA consistently outperforms baselines, suggesting that its design effectively handles score noise while still promoting semantic diversity. We will include above discussion in the revised version.
>
> ### **Q4: On Clustering Ablation and Design Choice**
>
> Thank you for raising this insightful question. We fully agree that isolating the effect of clustering is essential for understanding DORA’s improvements.
>
> In the early stages of our study, we explored several alternative clustering strategies before finalizing the soft clustering design:
>
> * **KMeans** clustering requires predefining the number of clusters \$K\$. However, the actual number of distinct reasoning directions can vary across problems and reasoning steps, making fixed \$K\$ a poor approximation and often leading to suboptimal clustering performance.
>
> * We also experimented with **hierarchical clustering**, which avoids setting \$K\$ by merging solutions based on a fixed cosine similarity threshold \$C\$. However, the optimal value of \$C\$ should ideally vary with the depth and complexity of reasoning, which differ significantly across problems. This sensitivity made hierarchical clustering even less effective than KMeans in our experiments.
>
> * Based on these findings, we adopted a **soft clustering** approach in DORA, which avoids hard assignment boundaries and allows for more adaptive and robust semantic grouping across diverse reasoning structures.
>
> We present below the ablation study comparing different clustering methods. All experiments were conducted on the MATH500 benchmark with rollout budget \$N=64\$ using the LLaMA-3.1-8B policy model (results averaged over 5 independent runs):
>
> |Cluster Method|K=3|K=5|K=10|C=0.95|C=0.90|C=0.80| Soft (default) |
> |--------------|----|----|----|------|------|------|--------------|
> |Accuracy| 67.6 |68.0|67.8|67.0|66.6|66.0|**68.7**|
>
> As shown, soft clustering consistently achieves the best performance, demonstrating its effectiveness over hard clustering alternatives. We will include this ablation and discussion in the revised version.
>
> ### **Q5: On Generalization Beyond Mathematical Reasoning**
>
> We appreciate the reviewer’s thoughtful question regarding DORA’s applicability beyond math reasoning. We fully agree that evaluating generalization across domains—such as code generation—is an important direction.
>
> In our current study, the primary obstacle to extending DORA to other domains is the lack of strong, open-source step-level Process Reward Models (PRMs) outside the math domain. Most high-quality PRMs publicly available today are specifically trained for mathematical reasoning \[4]. For the code generation domain, we had already conducted exploratory experiments early in our study using Skywork-PRM—the only open-source PRM we are aware of that provides step-level feedback for programming tasks. We applied it to standard benchmarks such as HumanEval and MBPP, and tested all step-level search methods (Beam Search, DVTS, REBASE, and DORA). However, none of these methods were able to significantly outperform the Temperature Sampling baseline.
>
> We believe this outcome is due to the limited step-level guidance capacity of Skywork-PRM for code tasks. This limitation has also been observed in recent work [5], which shows that Skywork-PRM provides relatively weak supervision for multi-step code generation. While using prompted open-source LLMs as critic models is a potential workaround, this approach introduces significant computational overhead and latency at inference time—undermining the efficiency goals of Test-Time Scaling.
>
> We will include this discussion in the revised version and acknowledge the limited domain coverage of current PRMs as a key limitation. Extending DORA to broader domains—such as code, science QA, or planning—remains an exciting and important direction that we leave for future work.
>
> [1] MTEB: Massive Text Embedding Benchmark, EACL2023.
>
> [2] Scaling Laws for Neural Language Models, OpenAI.
>
> [3] Scaling LLM Test-Time Compute Optimally Can be More Effective than Scaling Parameters for Reasoning, ICLR2025.
>
> [4] PROCESSBENCH: Identifying Process Errors in Mathematical Reasoning, ACL2025.
>
> [5] CodePRM: Execution Feedback-enhanced Process Reward Model for Code Generation, ACL2025.

---

> ### Author Response · Authors · 2025-08-06
>
> Dear Reviewer JSmU,
>
> We hope this message finds you well. As the discussion period is nearing its end, we would like to kindly follow up to ensure that our response has addressed your concerns satisfactorily. If there are any additional questions or suggestions you would like us to consider, please do not hesitate to let us know. Your insights are invaluable to us, and we would be more than happy to engage in further discussion to improve our work.
>
> Thank you once again for your time and thoughtful feedback！
>
> Best regards,
> The Authors

---

> ### Author Response · Authors · 2025-08-08
> **Gentle Reminder**
>
> Dear Reviewer JSmU,
>
> We are writing to you with a gentle follow-up as the discussion period will close in about 30 hours. We completely understand that your time is precious and you are likely very busy. We just wanted to kindly reiterate that if you have any final comments or concerns, we are fully prepared to do whatever is necessary to address them promptly.
>
> Thank you once again for your time and invaluable contribution to our work!
>
> Best regards,
>
> Authors

---

### Official Review · Reviewer_asTA · 2025-07-03

**Clarity:** 4
**Significance:** 3
**Originality:** 3
**Rating:** 5
**Confidence:** 3

**Summary:**

This paper tackles the problem of efficiently allocating a fixed computational budget during test-time search with Large Language Models. The authors reframe Test-Time Scaling as a constrained resource-allocation problem, providing a novel theoretical lens through which to analyze existing search strategies. Using a Bayesian framework with a Beta prior to model uncertainty in reward estimates, they derive the form of the optimal allocation. This analysis reveals that popular methods like Beam Search and REBASE are suboptimal because they allocate resources at the individual solution level, favoring "reasoning directions" that are overrepresented with many similar candidates.

The paper's core contribution is Direction-Oriented Resource Allocation (DORA), which computes semantic embeddings of candidate solutions to identify redundant paths, and then re-weights the rollout allocation to separate a direction's quality from the number of candidates representing it. Extensive experiments on challenging mathematical reasoning benchmarks show that DORA consistently achieves higher accuracy than strong baselines and requires 3.5x fewer FLOPs and 4x lower latency than the next-best method for comparable performance.

**Questions:**

This is an excellent submission, and my questions are mainly aimed at exploring the boundaries of the claims and the method's robustness. A strong response here could easily convince me to raise my score.

1. Exploring the Limits of Optimality: Your optimality proof hinges on some key assumptions. I'd be interested to hear more about what happens when they are pushed.
(i) How does DORA's performance hold up as the variance of PRM scores within a direction increases, violating the core assumption of Theorem 1?
(ii) What happens in the high-confidence regime $(k → \infty)$? When the PRM is highly reliable, the best strategy is to exploit the top candidate. Does DORA gracefully recover this behavior, or could its diversity-promoting mechanism become a liability by steering compute away from a single, truly optimal path?

2. Robustness to Hyperparameters: The performance seems to depend on a very specific temperature, $T_s = 0.01$. This value makes the softmax behave almost like a hard assignment. How sensitive is the method to this choice? An ablation plot showing performance as $T_s$ is varied (e.g., across {0.001, 0.01, 0.1, 1.0}) would be very helpful for understanding if the method is stable or requires careful tuning.

3. The Clustering Mechanism: DORA's effectiveness relies on its soft clustering. To better understand why it works, I have two questions:
(i) Have you considered a simpler "hard clustering" baseline? For example, one could run K-means on the embeddings and then allocate the budget at the cluster level. Comparing against this would clarify whether the benefit comes simply from accounting for redundancy, or if the soft-reweighting scheme is truly superior.
(ii) Could you provide a more direct, empirical measure of how well the clustering works? For instance, measuring the success-rate correlation between pairs of solutions before and after they are grouped by DORA would be a great way to quantify how effectively it identifies and mitigates dependencies.

**Ethical Concerns:**

["NO or VERY MINOR ethics concerns only"]

**Final Justification:**

I am maintaining my score.

**Limitations:**

Yes, the authors do a good job of discussing limitations. I'd encourage them to also briefly touch on the following points to make the section even stronger:
- The gap between the theoretical assumption of identical PRM scores and the practical reality.
- The introduction of new components (embedding model, T_s) and the potential for their selection to induce new biases or require careful tuning.
- A brief note that the empirical validation, while strong, is confined to mathematical reasoning, and generalization to other domains (e.g., code generation) remains an open and interesting question.

**Paper Formatting Concerns:**

Nothing egregious comes to mind

**Quality:**

4

**Strengths And Weaknesses:**

I found this to be a very strong paper, with its primary strengths being its principled theory, the novelty of the method, and a compelling set of experiments.

The paper’s theoretical grounding is a major strength. By moving from simple heuristics to a formal resource allocation problem, the authors offer a unified view of the test-time search landscape. This framework elegantly explains existing methods as special cases and provides a clear motivation for why they might fail. The proposed method, DORA, is a direct and clever solution to the problem identified by the theory. Using semantic similarity to compute a "uniqueness" score is a fresh approach that moves beyond simple diversity-promoting heuristics.

Most importantly, the method seems to work very well. The empirical results are a highlight. The gains in accuracy are consistent, but the efficiency gains are what truly make the contribution stand out. A 4x speedup, as shown in Table 1, is a significant practical achievement. The analysis is thorough, covering multiple models, benchmarks, and budget sizes, which gives me confidence in the results. The paper is also exceptionally well-written and easy to follow.

The weaknesses are minor in comparison but worth noting. The "provably optimal" claim rests on the simplifying assumption that all solutions within a "direction" have identical PRM scores, a condition that won't hold perfectly in practice. The paper doesn't explore how the guarantees degrade as this assumption is relaxed. Additionally, the method's performance is now tied to new components, the embedding model and a key temperature hyperparameter ($T_s$), whose robustness and sensitivity are not explored in an ablation study. While the chosen settings work well, understanding their stability is important for real-world deployment.

---

> ### Author Rebuttal · Authors · 2025-07-31
>
> Dear Reviewer asTA, we sincerely appreciate you taking the time to read our manuscript and provide valuable feedback. Below are our responses to the concerns you raised, which will be incorporated into the updated version to further enhance the quality of our submission.
>
> ### **Q1 & L1: On Exploring the Limits of Optimality**
>
> **(i) Robustness to Intra-Directional PRM Score Variance**
>
> Thank you for this thoughtful question. The assumption in Theorem 1—that all solutions in a direction are perfectly correlated with identical PRM scores—is made purely to simplify the theoretical analysis.
>
> In practice, solutions within a direction are usually partially correlated, and PRM scores can vary. Modeling this directly would make the theoretical formulation intractable. To handle this, DORA adopts a soft clustering design and retains per-candidate PRM scores $R_i$, without collapsing them at the direction level.
>
> Rollout allocation is computed as (line 195-196):
>
> $$w_i' = \frac{w_i \cdot \gamma_i}{\sum_j w_j \cdot \gamma_j},$$
>
> where $w_i$ reflects quality, and $\gamma_i$ accounts for semantic uniqueness. This enables DORA to prioritize better solutions even within noisy directions. Thus, while the theoretical assumption simplifies analysis, DORA’s implementation is robust to score variance and operates fully at the individual solution level. We will clarify this distinction in the revised version.
>
> **(ii) DORA’s Behavior in the High-Confidence Regime**
>
> Thank you for the insightful question. In the theoretical high-confidence case—where the PRM is perfectly reliable (\$\kappa \to \infty\$)—its scores exactly reflect the true success probabilities. For example, a PRM score of 0.9 means that sampling from that solution yields success with 90% probability. In this limit, the optimal strategy is indeed to allocate all rollouts to the top-scoring candidate.
>
> However, this regime is unrealistic in practice: computing true posterior success rates would require resampling each solution many times, which introduces significant cost and latency—contrary to the goal of test-time efficiency. Instead, PRMs in practice are noisy estimators trained on limited data, and this noise precludes direct exploitation.
>
> In such realistic settings, REBASE is optimal when assuming candidate independence, and DORA extends this by accounting for semantic dependencies, leading to more balanced allocations. Since DORA’s design assumes PRM noise and redundancy among solutions, it is inherently incompatible with the \$\kappa \to \infty\$ case, and thus does not recover greedy behavior in that limit.
>
> We will clarify this in the final version. Thank you again for encouraging this discussion.
>
> ### **Q2 & L2: On the Robustness to hyperparameter \$T\_s\$ and Embedding Model Choices**
>
> Thank you for this important question. We fully understand your concern regarding the robustness of DORA’s semantic grouping, which relies on the soft clustering temperature \$T\_s\$ and the embedding model.
>
> We first conducted an ablation study to evaluate the sensitivity of DORA to the semantic similarity temperature \$T\_s\$, which determines how close two solutions should be to be considered as belonging to the same reasoning direction. Experiments were performed on MATH500 with rollout budget \$N = 64\$, using LLaMA-3.1-8B-Instruct as the policy model. Results (averaged over 5 independent runs) are shown below:
>
> | \$T\_s\$ |0.001|0.01|0.1|1.0|
> | -------- | ----- | ---- | ---- | ---- |
> | DORA     | 67.8  | 68.7 | 68.0 | 67.5 |
>
> The results suggest that DORA is relatively robust to variations in \$T\_s\$.
>
> To further assess the impact of the embedding model itself, we compared our original retriever (BGE-M3, 568M parameters) with two popular alternatives from the MTEB leaderboard \[1] that support long input lengths (≥2048 tokens): E5-base-4k (110M) and GTE-multilingual-base (305M). All experiments were conducted on MATH500 with \$N = 64\$, under the same experimental settings. Results are averaged over 5 runs:
>
> | Policy Model|BGE|GTE|E5|
> | ------------- | ---- | ---- | ---- |
> | LLaMA-3.2-1B  | 68.7 | 68.2 | 66.8 |
> | LLaMA-3.2-3B  | 77.6 | 77.4 | 76.8 |
> | Qwen-2.5-1.5B | 80.8 | 80.2 | 79.2 |
>
> We observe that GTE and BGE perform comparably, showing that DORA is robust to embedding model changes within a similar capacity range. E5 does show a modest drop in performance, which we attribute to its smaller size and hence weaker ability to cluster semantically meaningful reasoning chains.
>
> We will include these findings and discussions in the final version to better address concerns about DORA’s sensitivity to hyperparameter and embedding choices.
>
> ### **Q3: On the Clustering Mechanism**
>
> **(i) Comparison to Hard Clustering Baselines**
>
> Thank you for raising this insightful question. We fully agree that understanding the role of clustering is key to explaining DORA’s effectiveness.
>
> In the early stages of our study, we systematically explored several hard clustering alternatives before adopting the final soft clustering design:
>
> * **KMeans clustering** requires predefining the number of clusters \$K\$. However, the actual number of distinct reasoning directions varies significantly across problems and reasoning steps. Using a fixed \$K\$ often led to poor alignment with the true semantic structure, resulting in suboptimal cluster quality.
>
> * We also considered **hierarchical clustering**, which avoids setting \$K\$ by merging solutions based on a fixed cosine similarity threshold \$C\$. However, we found that the optimal value of \$C\$ should ideally vary with the reasoning depth and problem type—factors that are highly instance-dependent. Consequently, hierarchical clustering was even less stable than KMeans in our experiments.
>
> Based on these observations, we adopted a **soft clustering** approach in DORA, which avoids hard assignment boundaries and allows for more adaptive semantic grouping. Below is our ablation comparing different clustering choices, conducted on the MATH500 benchmark with rollout budget \$N=64\$ using LLaMA-3.1-8B as the policy model (results are averaged over 5 independent runs):
>
> | Cluster Method | K=3  | K=5  | K=10 | C=0.95 | C=0.90 | C=0.80 | Soft (default) |
> | -------------- | ---- | ---- | ---- | ------ | ------ | ------ | -------------- |
> |Accuracy|67.6| 68.0|67.8|67.0| 66.6|66.0|**68.7**|
>
> As shown, soft clustering consistently outperforms hard clustering baselines, highlighting the importance of flexible, non-discrete grouping in capturing subtle reasoning structures. We will include this ablation study and analysis in the updated version.
>
> **(ii) Empirical Effectiveness of Semantic Grouping**
>
> We appreciate the reviewer’s thoughtful suggestion to quantify how effectively DORA’s clustering mitigates redundancy and improves success rates. This is indeed a highly valuable diagnostic perspective.
>
> To better understand the effect of DORA’s semantic grouping mechanism, we conducted an additional analysis focusing on how well each method improves the intermediate success rate along the reasoning trajectory. Specifically, after each method allocates $K$ reasoning steps, we remove the search algorithm and resume temperature sampling based on the intermediate solutions obtained thus far. We then measure the pass rate of these partial trajectories to estimate the success rate at step $K$. This reflects the method’s ability to guide the policy model toward more promising reasoning directions early on.
>
> Experiments were conducted on the MATH500 benchmark with rollout budget $N=64$, using LLaMA-3.1-8B-Instruct as the policy model and Qwen2.5-Math-PRM-7B as the PRM. All results were averaged over 5 independent runs:
>
> | Step|0|5|10|15|20|25|30|35|40|
> | ----------- | ---- | ---- | ---- | ---- | ---- | ---- | ---- | ---- | ----|
> |Beam Search| 27.7 | 39.3 | 49.6 | 54.2 | 55.8 | 56.8| 57.3|57.5|57.5|
> |DVTS| 27.7 | 36.5 | 40.8 | 41.7 | 41.9 | 41.9 | 41.9|41.9|42.1|
> | REBASE| 27.7 | 39.5 | 51.2 | 54.5 | 56.5 | 57.1|58.0|58.3|58.3|
> | DORA| 27.7|**40.2**|**51.6**|**55.6**|**57.4**| **58.2** | **59.0** | **59.5** | **59.6** |
>
> As shown, DORA consistently achieves the highest pass rates across intermediate steps compared to all baselines. Notably, Step 0 corresponds to the Temperature Sampling baseline (i.e., without any search intervention). Comparing this with the improvements achieved by REBASE and DORA highlights the value of clustering: while both methods significantly outperform the baseline, DORA consistently maintains a lead, suggesting that its semantic clustering mechanism not only reduces redundancy but also enhances the effectiveness of search guidance. We will include this result and analysis in the revised version.
>
> ### **L3: On Generalization Beyond Mathematical Reasoning**
>
> Thank you for pointing out this important limitation. We fully agree that evaluating generalization across domains is an important direction.
>
> DORA’s generalization beyond math (e.g., to code) is limited mainly by the lack of open-source step-level PRMs outside the math domain [2]. We have tested DORA and other step-level methods (Beam Search, DVTS, REBASE) on HumanEval and MBPP using Skywork-PRM—the only available PRM for programming tasks—but none significantly outperformed Temperature Sampling. This aligns with recent findings [3] showing Skywork-PRM offers weak step-level supervision. Using LLMs as critics is a potential alternative but adds considerable inference-time overhead, conflicting with TTS efficiency goals.
>
> We will include this discussion and highlight the domain limitation. Extending DORA to code, science QA, and planning is a promising direction for future work.
>
> [1] MTEB: Massive Text Embedding Benchmark, EACL2023.
>
> [2] PROCESSBENCH: Identifying Process Errors in Mathematical Reasoning, ACL2025.
>
> [3] CodePRM: Execution Feedback-enhanced Process Reward Model for Code Generation, ACL2025.

---

> ### Author Response · Authors · 2025-08-06
>
> Dear Reviewer asTA,
>
> We hope this message finds you well. As the discussion period is nearing its end, we would like to kindly follow up to ensure that our response has addressed your concerns satisfactorily. If there are any additional questions or suggestions you would like us to consider, please do not hesitate to let us know. Your insights are invaluable to us, and we would be more than happy to engage in further discussion to improve our work.
>
> Thank you once again for your time and thoughtful feedback！
>
> Best regards,
> The Authors

---

> ### Author Response · Authors · 2025-08-08
> **Gentle Reminder**
>
> Dear Reviewer asTA,
>
> We are writing to you with a gentle follow-up as the discussion period will close in about 30 hours. We completely understand that your time is precious and you are likely very busy. We just wanted to kindly reiterate that if you have any final comments or concerns, we are fully prepared to do whatever is necessary to address them promptly.
>
> Thank you once again for your time and invaluable contribution to our work!
>
> Best regards,
>
> Authors

---

### Official Review · Reviewer_N3ut · 2025-07-03

**Clarity:** 2
**Significance:** 3
**Originality:** 3
**Rating:** 5
**Confidence:** 3

**Summary:**

The paper presents a critical analysis of rollout allocation efficiency in Test-Time Scaling for Large Language Models (LLMs) focused on mathematical reasoning, leveraging Process Reward Models (PRMs). It identifies limitations in prior methods by demonstrating that the assumption of independence between rollout allocations fails in practical scenarios, leading to suboptimal performance. To address this, the authors propose a theoretical framework that incorporates embedding similarity scores to mitigate the breakdown of independence, thereby achieving optimality under the proposed conditions. The framework is rigorously evaluated on three standard mathematical reasoning benchmarks using a 7B-parameter PRM and 1 to 3B-parameter generative models, demonstrating its effectiveness in improving scalability and performance for mathematical tasks. The work contributes a novel approach to optimizing test-time scaling in LLMs through both theoretical justification and empirical validation.

**Questions:**

1. How much sensitive could the proposed method be towards the choice of retriever (E5/GTE instead of  BGE)?
2. Which chat template is used for Llama models as  there seems to be an issue related to it[1]?
3. The quality score parameter T_b is not well ablated in the REBASE paper. How much are the results in the proposed paper sensitive to this parameter?


References:

1. https://huggingface.co/spaces/HuggingFaceH4/blogpost-scaling-test-time-compute

**Ethical Concerns:**

["NO or VERY MINOR ethics concerns only"]

**Final Justification:**

All the major concerns have been addressed:
- More benchmarks to demonstrate generalization.
- Ablated hyperparameters related to temperature.
- Discussed potential practical bottlenecks.
- The method shines irrespective of the choice of retriever.

**Limitations:**

Yes

**Quality:**

4

**Strengths And Weaknesses:**

### Strengths
1. Theoretical Justification: The method derives the sub-optimal conditions for REBASE and proposes more effective method based on an additional score from embedding similarity. The theoretical result establishes the optimality of the direction-level allocation schema proposed in the paper.

2. Performance: The proposed framework achieves superior performance at lower computer cost in comparison to the baseline, for example, 3.5 times  fewer flop in MATH500, resulting in a lower latency as well.

### Weaknesses
1. The generalization of the presented results are limited. First, the  improvement is mainly visible for MATH500, but for AIME task, there is a good amount of overlap with baselines for statistical significance. Next, the range of policy models are limited as it covers only 1-3B models. Evaluating larger model classes such as 8B and 32B could further establish the significance of the proposed methods. Similarly, only single PRM is evaluated. The results for larger PRMs (72B) and other families (e.g., Skywork) could clarify the scalability of the  proposed method.

2. Motivation of  some design choices are  unclear. For example, semantic similarity temperature is chosen as 0.01. It is unclear how is it tuned and the sensitivity of the results due  to this parameter.

3. The proposed method introduces an additional model dependency for Test-Time Scaling which could hinder its future extension.  For example, it the retriever model is not  trained with code data, the scaling method will  face bottleneck for coding benchmarks.

4. Table 1 does not include the error bars (deviation) and statistics for AIME tasks. Including these metrics could further clarify the scalability of the proposed framework.

---

> ### Author Rebuttal · Authors · 2025-07-30
>
> Dear Reviewer N3ut, we sincerely appreciate you taking the time to read our manuscript and provide valuable feedback. Below are our responses to the concerns you raised, which will be incorporated into the updated version to further enhance the quality of our submission.
>
> ### **W1: Regarding the Generalization of the Presented Results**
>
> We understand your concern and agree with your suggestion that conducting experiments on larger policy and PRM models, and across different PRM families would better validate the generalizability of DORA. Regarding the AIME tasks, we note that the observed overlap with baseline performance is primarily due to the small dataset sizes of AIME24 and AIME25 (only 30 problems each), which makes the Accuracy metric more prone to high variance and less statistically stable. To enhance the statistical reliability of the results, we followed [1] and conducted 10 independent runs with different random seeds, reporting the averaged performance (see lines 214–215). To further demonstrate the generality and scalability of DORA, we conducted the following three sets of additional experiments (All results are averaged over 5 independent runs to ensure statistical stability):
>
> (1) Broader Benchmark Coverage: We incorporated four additional standard math reasoning benchmarks: HMMT24, HMMT25, AMC23, and AMC24. Under a rollout budget of N=64, the results are summarized as follows:
>
> |Policy Model|Method|HMMT24|HMMT25|AMC23|AMC24|
> |------------|------|--------|--------|--------|--------|
> |LLaMA-3.2-1B|Temperature Sampling |0.0|0.0|28.0|13.3|
> ||Beam Search |2.0     |0.0     |43.0|15.5|
> ||DVTS  |1.3     |0.0     |41.0    |18.0|
> ||REBASE|2.0     |0.0     |43.0    |18.2 |
> ||DORA  |**3.3** |0.0 |**45.0**|**20.0**|
> |LLaMA-3.2-3B|Temperature Sampling |0.7|0.0|48.5    |21.7    |
> ||Beam Search  |2.0     |0.7     |55.5 |25.3    |
> ||DVTS  |2.0     |0.0     |57.0    |26.2    |
> ||REBASE|4.7     |0.0     |**59.0**|27.5    |
> ||DORA  |**6.7** |**2.0** |**59.0**|**31.1**|
> |Qwen-2.5-1.5B|Temperature Sampling   |4.0     |0.0     |42.0    |16.4    |
> ||Beam Search  |4.7     |1.3     |53.0    |31.1    |
> ||DVTS  |4.7     |2.0     |51.5    |24.9    |
> ||REBASE|5.3     |2.7     |54.0    |32.0    |
> ||DORA  |**6.7** |**4.0** |**55.0**|**34.2**|
>
>
> (2) Larger Policy Models and PRM: We evaluated DORA with more powerful models by using Qwen2.5-Math-PRM-72B as the PRM to guide the LLaMA-3.1-8B-Instruct policy model on the MATH500 benchmark with N=64. The performance is reported below:
>
> |PRM |Policy Model|Temperature Sampling|Beam Search|DVTS|REBASE|DORA|
> |-----|-----|----|----|-----|------|-----|
> |Qwen2.5-Math-PRM-72B|LLaMA-3.1-8B|70.5|78.2|78.4|79.2|**80.6**|
>
> (3) Different PRM Families: To evaluate generalization across PRM families, we additionally tested **Skywork-PRM-7B** on the MATH500 benchmark with N=64. The results are shown below:
>
> |Method|LLaMA-3.2-1B|LLaMA-3.2-3B|Qwen-2.5-1.5B|
> |------|------------|------------|-------------|
> |Temperature Sampling|58.5|69.5|73.0|
> |Beam Search|69.0|75.5|79.7|
> |DVTS|66.2|74.2|78.9|
> |REBASE|70.8|76.2|80.2|
> |DORA|**71.8**|**76.6**|**81.0**|
>
> Above experiments demonstrate that DORA generalizes well across a broader range of benchmarks, scales to larger policy models and PRMs, and is compatible with PRMs from different families. Specifically, the experiments with Qwen2.5-Math-PRM-72B were executed in parallel on a cluster with 100 H20 GPUs (96GB each), using 2 GPUs per run, and were completed in approximately 4 days. We sincerely thank you for this insightful suggestion and will include the results and analysis in the revised version.
>
> ### **W2 & Q3: On the Design Choice of Hyperparameters $T_s$ and $T_b$**
>
> We fully understand your concern regarding the motivation and sensitivity of certain hyperparameters. For the quality score temperature $T_b$, we default to $T_b=0.1$ to align with REBASE, ensuring a fair and consistent comparison. For the semantic similarity temperature $T_s$, which determines how close two solutions should be to be considered as belonging to the same reasoning direction, we conducted an ablation study to select a suitable value. We apologize for the oversight in not including this ablation result in the original submission. We performed a detailed sensitivity analysis of both $T_s$ and $T_b$ on the MATH500 benchmark with rollout budget N=64, using LLaMA-3.1-8B-Instruct as the policy model. All results are averaged over 5 independent runs to ensure statistical stability. The results are summarized as follows:
>
> |$T_b$   |0.01|0.1|1.0|
> |------|----|----|----|
> |REBASE|64.8|65.9|55.4|
> |DORA  |67.4|68.7|57.2|
>
> |$T_s$  |0.001|0.01|0.1|1.0|
> |-----|-----|----|----|----|
> |DORA|67.8 |68.7|68.0|67.5|
>
> These results show that DORA is relatively robust to the choice of $T_s$. As for $T_b$, both REBASE and DORA achieve strong performance when $Tb=0.01$ or $0.1$, but exhibit a clear performance drop when $T_b=1.0$. This degradation occurs because a high temperature in the softmax weakens the guidance of the PRM. In the extreme case where $T_b \to \infty$, REBASE degenerates into temperature sampling without PRM guidance.  We will include these results and discussion in the later version.
>
> ### **W3: About Potential Bottlenecks for Test-Time Scaling**
>
> We understand and appreciate your concern regarding the additional model dependency introduced by Test-Time Scaling (TTS), particularly the reliance on an embedding model for retrieval. We acknowledge that if the embedding model is trained on data with limited coverage of a target domain, its ability to cluster reasoning chains in that domain could indeed be suboptimal. This is a valid concern when considering the broader applicability of TTS.
>
> However, recent advances in embedding model development have increasingly moved toward general text embeddings, as demonstrated in [2]. Modern retrievers such as GTE-Qwen and E5-Mistral are designed to inherit rich linguistic and domain knowledge from large pre-trained language models (e.g., Qwen2-1.5B), which cover a wide range of reasoning and generation scenarios. Based on this trend, we believe that the inclusion of an embedding model in TTS is unlikely to become a bottleneck for extending TTS to other domains, as many embedding models today possess sufficient generalization capabilities to support effective clustering across diverse tasks.
>
> Respectfully, in our view, the more significant limitation to extending TTS beyond math currently lies in the lack of high-quality, open-source PRMs for domains such as code generation. Most existing PRMs are tailored for math reasoning [3], while open-source PRMs for code tasks remain scarce. To the best of our knowledge, Skywork-PRM is among the few publicly available options, but recent work [4] has shown that it offers relatively weak step-level supervision on code-related tasks. While one alternative is to use open-source language models prompted as critic models, this introduces substantial computational overhead and time delay at test time, which undermines the efficiency advantages of TTS.
>
>
> ### **W4: About Including Error Bars and Statistics for AIME on Table 1**
>
> We sincerely thank the reviewer for this helpful suggestion. The completed results are shown below, which will be incorporated into the later version:
>
> |Dataset|Method|Rollout|FLOPS|Latency(s)|Accuracy|
> |--------|------------|-------|----------------------|----------|-----------|
> |MATH500 |Beam Search |256|2.86(0.03)×10¹⁵ |345(7) |63.6(0.8) |
> ||DVTS |256|3.03(0.03)×10¹⁵ |253(8)    |62.0(0.9)  |
> ||REBASE|256    |3.11(0.03)×10¹⁵ |490(10)    |67.4(0.8)  |
> ||DORA|64     |8.92(0.05)×10¹⁴ |124(8)    |68.7(0.8)|
> |AIME24|Beam Search |256|5.67(0.17)×10¹⁵|816(16)         |11.3(2.8) |
> ||DVTS|256    |3.99(0.05)×10¹⁵  |734(9)         |11.6(2.4)  |
> ||REBASE|256    |5.67(0.05)×10¹⁵ |978(14)         |14.7(2.3) |
> ||DORA|64     |1.72(0.19)×10¹⁵|240(10)         |14.7(2.3)|
>
>
>
>
> ### **Q1: About sensitive of different retrievers**
>
> We understand your concern about the potential sensitivity of DORA to different retriever models. To further investigate DORA’s robustness across retriever families, we compared our original retriever (bge-m3, 568M parameters) with two popular alternatives from the MTEB leaderboard [5] that support long inputs (≥2048 tokens): e5-base-4k (110M) and gte-multilingual-base (305M).All experiments were conducted on MATH500 with N=64, and all other settings were kept consistent with the main experiments. All results are averaged over 5 independent runs to ensure statistical stability. The results are summarized below:
>
> |Policy Model|bge |gte|e5|
> |-----------------|-----|-----|-----|
> |LLaMA-3.2-1B |68.7 |68.2 |66.8 |
> |LLaMA-3.2-3B |77.6 |77.4 |76.8 |
> |Qwen-2.5-1.5B |80.8 |80.2 |79.2 |
>
>
> The results show that GTE and BGE yield comparable performance, indicating that DORA is robust to retriever changes within similar capacity ranges. However, E5 exhibits certain performance degradation, which we hypothesize is due to its smaller model size and thus weaker clustering capability in high-dimensional embedding space.
> We will include these results and analysis in the revised version to clarify DORA’s sensitivity to retriever choices.
>
> ### **Q2: About chat template used for Llama models**
>
> We confirm that the chat template used for LLaMA models in our experiments is exactly aligned with the configuration described in [6] (see Appendix D.2). In fact, our codebase is directly built upon the official open-source implementation provided by [6].
>
> [1] A sober look at progress in language model reasoning: Pitfalls and paths to reproducibility.
>
> [2] Towards General Text Embeddings with Multi-Stage Contrastive Learning.
>
> [3] PROCESSBENCH: Identifying Process Errors in Mathematical Reasoning, ACL2025.
>
> [4] CodePRM: Execution Feedback-enhanced Process Reward Model for Code Generation, ACL2025.
>
> [5] MTEB: Massive Text Embedding Benchmark, EACL2023.
>
> [6] blogpost-scaling-test-time-compute, HuggingFace.

---

> > ### Comment · Reviewer_N3ut · 2025-08-04
> >
> > I thank the authors for their detailed response and experiments. As most of my concerns are addressed, I am willing to increase the rating by 2 points.

---

> > > ### Author Response · Authors · 2025-08-04
> > > **Reply to Reviewer N3ut**
> > >
> > > Thank you for your timely reply and your recognition of our work! We will make sure to include the above discussion in the final version of the paper. Once again, we would like to express our heartfelt appreciation for the time and effort you dedicated to reviewing our submission. Your insightful suggestions are undoubtedly invaluable, and we are confident that they will greatly improve the quality of our work.

---

### Author Response · Authors · 2025-08-09
**Final Comment**

We sincerely thank all reviewers for dedicating their valuable time to reviewing our manuscript, as well as for recognizing our work and providing helpful suggestions. We are pleased to note that the reviewers generally acknowledge our strengths:

* **Theoretical Grounding [N3ut, asTA, T3Po]:** The study reframes test-time scaling as a principled resource allocation problem, deriving optimal strategies via Bayesian probabilistic modeling. This unified theoretical lens offers a principled explanation of the limitations of existing methods, subsumes them as special cases, and rigorously establishes the optimality of the proposed DORA schema.

* **Clear Novelty [N3ut, asTA, JSmU]:** The proposed DORA method leverages semantic embedding similarity to decouple a direction’s quality from its candidate count, moving beyond diversity heuristics to address an underexplored aspect of rollout allocation.

* **Effectiveness of the proposed DORA method [N3ut, asTA, JSmU, T3Po]:** Extensive experiments on challenging benchmarks show that DORA consistently outperforms strong baselines while achieving up to 3.5× fewer FLOPs and 4× lower latency across various models and budgets.

* **Sound analysis [asTA]:** The analysis is thorough, covering multiple models, benchmarks, and budget sizes.

* **Good readability [asTA, T3Po]:** The paper is exceptionally well-written and easy to follow.

---
We also deeply appreciate the reviewers’ detailed suggestions from different perspectives, which we have addressed one by one during the rebuttal phase. These resolutions will be incorporated into the revised version, which we believe will further enhance the quality of our manuscript:

* **Testing DORA’s generalizability on broader settings and domains [N3ut, T3Po, asTA, JSmU]:** Our experiments during the discussion phase confirmed that DORA generalizes well across a broader range of benchmarks, scales to larger policy models and PRMs, and is compatible with PRMs from different families. Including these results in the paper will strengthen the credibility and generalization capability of DORA.

* **Illustrating sensitivity and choice of hyperparameters [N3ut, asTA, JSmU, T3Po]:** We further evaluated DORA under multiple hyperparameter settings, including $T_s$, $T_b$, and retrievers, and the results validated DORA’s robustness to multiple hyperparameter choices.

* **Discussing the theoretical assumption of shared PRM scores within a direction [asTA, JSmU]:** We clarified that the shared-score assumption was introduced solely for tractable analysis. In practice, DORA applies soft clustering with per-candidate PRM scores to handle intra-directional variance, ensuring robustness under this setting. Including this discussion will make the motivation of our method easier to understand.

* **Analysing potential computational bottlenecks for Test-Time Scaling [N3ut, T3Po]:** Following reviewers’ suggestion, we provided a formula-based analysis showing that the embedding computation cost of DORA does not become a computational bottleneck as the TTS rollout budget $N$ increases.

* **Explaining the choice of clustering method [asTA, JSmU]:** Following reviewers’ suggestion, we clarified the design choice of using soft clustering in DORA and demonstrated through experiments that it outperforms hard clustering methods such as K-means and hierarchical clustering. Furthermore, we experimentally confirmed that DORA’s semantic grouping indeed improves the success rate of search.

* **More details [N3ut]:** Based on reviewer N3ut’s suggestion, we included error bars and statistics for AIME in Table 1, which further clarify the scalability of the proposed method.

---

We are glad that, after our rebuttal, Reviewer N3ut and Reviewer T3Po confirmed that most of their concerns have been addressed. We regret that we did not receive further responses from Reviewer asTA and Reviewer JSmU, and thus missed the opportunity to continue an in-depth discussion with these insightful and constructive reviewers. We would still very much value their feedback, even though less than two hours remain before the author–reviewer discussion phase ends. Once again, we sincerely thank all reviewers for dedicating their time and effort to reviewing our manuscript and for providing thoughtful and valuable suggestions. We respectfully leave the floor to the esteemed Area Chair and reviewers for the next stage of the discussion.

---

### Decision · Program_Chairs · 2025-09-17

**Decision:**

Accept (poster)

**Comment:**

Basically, this paper is built upon Reward Balanced Search (REBASE), and the research point is to arguing the assumption of REBASE that candidate independence. Based upon this arguement, the authors proposed DORA that introduces semantic embedding similarity to decouple rollout direction quality from candidate count, moving beyond diversity heuristics.
The idea is interesting and sound, the work is also with solid theoretical analysis.
After the discussion between reviewers, all reviewers are lean to accept.
I think the work is sound with solid contribution, thus champion the acceptance.
Here are some suggestions for the final version:
1). include the results and revision in the discussion to the final camera-ready;
2). include discussion of these two papers as they are highly relevant:
a). TREEBON: ENHANCING INFERENCE-TIME ALIGNMENT WITHSPECULATIVE TREE-SEARCH AND BEST-OF-N SAMPLING, Mendi Wang et. al. https://arxiv.org/pdf/2410.16033 (This prunes braches using PRM)
b). DISC: Dynamic Decomposition Improves LLM Inference Scaling. https://openreview.net/forum?id=FFl6iyvJX2 (This also dynamically allocate resources by using ORM, thus is also doable to closed-sourced LLMs).